# MALIGN OVERFITTING: INTERPOLATION CAN PROVABLY PRECLUDE INVARIANCE

**Yoav Wald**
Johns Hopkins University
ywald1@jhu.edu

**Gal Yona**
Weizmann Institute of Science

**Uri Shalit**
Technion

**Yair Carmon**
Tel Aviv University

## ABSTRACT

Learned classifiers should often possess certain invariance properties meant to encourage fairness, robustness, or out-of-distribution generalization. However, multiple recent works empirically demonstrate that common invariance-inducing regularizers are ineffective in the over-parameterized regime, in which classifiers perfectly fit (i.e. interpolate) the training data. This suggests that the phenomenon of "benign overfitting," in which models generalize well despite interpolating, might not favorably extend to settings in which robustness or fairness are desirable.

In this work, we provide a theoretical justification for these observations. We prove that—even in the simplest of settings—any interpolating learning rule (with an arbitrarily small margin) will not satisfy these invariance properties. We then propose and analyze an algorithm that—in the same setting—successfully learns a non-interpolating classifier that is provably invariant. We validate our theoretical observations on simulated data and the Waterbirds dataset.

## 1 INTRODUCTION

Modern machine learning applications often call for models which are not only accurate, but which are also robust to distribution shifts or satisfy fairness constraints. For example, we might wish to avoid using hospital-specific traces in X-ray images (DeGrave et al., 2021; Zech et al., 2018), as they rely on spurious correlations that will not generalize to a new hospital, or we might seek "Equal Opportunity" models attaining similar error rates across protected demographic groups, e.g., in the context of loan applications (Byanjankar et al., 2015; Hardt et al., 2016). A developing paradigm for fulfilling such requirements is learning models that satisfy some notion of *invariance* (Peters et al., 2016; 2017) across environments or sub-populations. For example, in the X-ray case, spurious correlations can be formalized as relationships between a feature and a label which vary across hospitals (Zech et al., 2018). Equal Opportunity (Hardt et al., 2016) can be expressed as a statistical constraint on the outputs of the model, where the false negative rate is invariant to membership in a protected group. Many techniques for learning invariant models have been proposed including penalties that encourage invariance (Arjovsky et al., 2019; Krueger et al., 2021; Veitch et al., 2021; Wald et al., 2021; Puli et al., 2021; Makar et al., 2022; Rame et al., 2022; Kaur et al., 2022), data re-weighting (Sagawa et al., 2020a; Wang et al., 2021; Idrissi et al., 2022), causal graph analysis (Subbaswamy et al., 2019; 2022), and more (Ahuja et al., 2020).

While the invariance paradigm holds promise for delivering robust and fair models, many current invariance-inducing methods often fail to improve over naive approaches. This is especially noticeable when these methods are used with overparameterized deep models capable of *interpolating*, i.e., perfectly fitting the training data (Gulrajani & Lopez-Paz, 2021; Dranker et al., 2021; Guo et al., 2022; Zhou et al., 2022; Menon et al., 2021; Veldanda et al., 2022; Cherepanova et al., 2021). Existing theory explains why overparameterization hurts invariance for standard interpolating learning rules, such as empirical risk minimization and max-margin classification (Sagawa et al., 2020b; Nagarajan et al., 2021; D'Amour et al., 2022), and also why reweighting and some types of distributionally robust optimization face challenges when used with overparameterized models (Byrd & Lipton, 2019; Sagawa et al., 2020a). In contrast, training overparameterized models to interpolate the training data typically results in good *in-distribution* generalization, and such "benign overfitting" (Kini et al., 2021; Wang et al., 2021) is considered a key characteristic of modern deep learning (Cao et al., 2021; Wang & Thrampoulidis, 2021; Shamir, 2022). Consequently, a num-

ber of works attempt to extend benign overfitting to robust or fair generalization by designing new interpolating learning rules (Cao et al., 2019; Kini et al., 2021; Wang et al., 2021; Lu et al., 2022).

In this paper, we demonstrate that such attempts face a fundamental obstacle, because *all* interpolating learning rules (and not just maximum-margin classifiers) fail to produce invariant models in certain high-dimensional settings where invariant learning (without interpolation) is possible. This *does not* occur because there are no invariant models that separate the data, but because interpolating learning rules *cannot find them*. In other words, beyond identically-distributed test sets, overfitting is no longer benign. More concretely, we consider linear classification in a basic overparameterized Gaussian mixture model with invariant "core" features as well as environment-dependent "spurious" features, similar to models used in previous work to gain insight into robustness and invariance (Schmidt et al., 2018; Rosenfeld et al., 2021; Sagawa et al., 2020b). We show that any learning rule producing a classifier that separates the data with non-zero margin must necessarily rely on the spurious features in the data, and therefore cannot be invariant. Moreover, in the same setting we analyze a simple two-stage algorithm that can find accurate and nearly invariant linear classifiers, i.e., with almost no dependence on the spurious feature.

Thus, we establish a separation between the level of invariance attained by interpolating and non-interpolating learning rules. We believe that learning rules which fail in the simple overparameterized linear classification setting we consider are not likely to succeed in more complicated, real-world settings. Therefore, our analysis provides useful guidance for future research into robust and fair machine learning models, as well as theoretical support for the recent success of non-interpolating robust learning schemes (Rosenfeld et al., 2022; Veldanda et al., 2022; Kirichenko et al., 2022; Menon et al., 2021; Kumar et al., 2022; Zhang et al., 2022; Idrissi et al., 2022; Chatterji et al., 2022).

**Paper organization.** The next section formally states our full result (Theorem 1). In Section 3 we outline the arguments leading to the negative part of Theorem 1, i.e., the failure of interpolating classifiers to be invariant in our model. In Section 4 we establish the positive part Theorem 1, by providing and analyzing a non-interpolating algorithm that, in our model, achieves low robust error. We validate our theoretical findings with simulations and experiments on the Waterbirds dataset in Section 5, and conclude with a discussion of additional related results and directions for future research in Section 6.

## 2 STATEMENT OF MAIN RESULT

### 2.1 PRELIMINARIES

**Data model.** Our analysis focuses on learning linear models over covariates $\mathbf{x}$ distributed as a mixture of two Gaussian distributions corresponding to the label $y$.

**Definition 1.** *An* environment *is a distribution parameterized by* $(\boldsymbol{\mu}_c, \boldsymbol{\mu}_s, d, \sigma, \theta)$ *where* $\theta \in [-1, 1]$ *and* $\boldsymbol{\mu}_c, \boldsymbol{\mu}_s \in \mathbb{R}^d$ *satisfy* $\boldsymbol{\mu}_c \perp \boldsymbol{\mu}_s$ *and with samples generated according to:*

$$\mathbb{P}_\theta(y) = \mathrm{Unif}\{-1, 1\}, \quad \mathbb{P}_\theta(\mathbf{x}|y) = \mathcal{N}(y\boldsymbol{\mu}_c + y\theta\boldsymbol{\mu}_s, \sigma^2 I). \tag{1}$$

Our goal is to find a (linear) classifier that predicts $y$ from $\mathbf{x}$ and is robust to the value of $\theta$ (we discuss the specific robustness metric below). To do so, the classifier will need to have significant inner product with the "core" signal component $\boldsymbol{\mu}_c$ and be approximately orthogonal to the "spurious" component $\boldsymbol{\mu}_s$. We focus on learning problems where we are given access to samples from two environments that share all their parameters other than $\theta$, as we define next. We illustrate our setting with Figure 3 in Appendix A.

**Definition 2** (Linear Two Environment Problem). *In a Linear Two Environment Problem we have datasets* $S_1 = \{\mathbf{x}_i^{(1)}, y_i^{(1)}\}_{i=1}^{N_1}$ *and* $S_2 = \{\mathbf{x}_i^{(2)}, y_i^{(2)}\}_{i=1}^{N_2}$ *of sizes* $N_1, N_2$ *drawn from* $\mathbb{P}_{\theta_1}$ *and* $\mathbb{P}_{\theta_2}$ *respectively. A learning algorithm is a (possibly randomized) mapping from the tuple* $(S_1, S_2)$ *to a linear classifier* $\mathbf{w} \in \mathbb{R}^d$. *We let* $S = \{\mathbf{x}_i, y_i\}_{i=1}^{N}$ *denote that dataset pooled from* $S_1$ *and* $S_2$ *where* $N = N_1 + N_2$. *Finally we let* $r_c := \|\boldsymbol{\mu}_c\|$ *and* $r_s := \|\boldsymbol{\mu}_s\|$.

We study settings where $\theta_1, \theta_2$ are fixed and $d$ is large compared to $N$, i.e. the overparameterized regime. We refer to the two distributions $\mathbb{P}_{\theta_e}$ for $e \in \{1, 2\}$ as "training environments", following Peters et al. (2016); Arjovsky et al. (2019). In the context of Out-of-Distribution (OOD) generalization, environments correspond to different experimental conditions, e.g., collection of medical data

in two hospitals. In a fairness context, we may think of these distributions as subpopulations (e.g., demographic groups).[1]While these are different applications that require specialized methods, the underlying formalism of solutions is often similar (see, e.g., Creager et al., 2021, Table 1), where we wish to learn a classifier that in one way or another is invariant to the environment variable.

**Robust performance metric.** An advantage of the simple model defined above is that many of the common invariance criteria all boil down to the same mathematical constraint: learning a classifier that is orthogonal to $\boldsymbol{\mu}_s$, which induces a spurious correlation between the environment and the label. These include Equalized Odds (Hardt et al., 2016), conditional distribution matching Li et al. (2018), calibration on multiple subsets of the data (Hébert-Johnson et al., 2018; Wald et al., 2021), Risk Extrapolation (Krueger et al., 2021) and CVaR fairness (Williamson & Menon, 2019).

In terms of predictive accuracy, the goal of learning a linear model that aligns with $\boldsymbol{\mu}_c$ (the invariant part of the data generating process for the label) and is orthogonal to $\boldsymbol{\mu}_s$ coincides with providing guarantees on the robust error, i.e. the error when data is generated with values of $\theta$ that are different from the $\theta_1, \theta_2$ used to generate training data.[2]

**Definition 3** (Robust error). *The robust error of a linear classifier* $\mathbf{w} \in \mathbb{R}^d$ *is:*

$$\max_{\theta \in [-1,1]} \epsilon_\theta(\mathbf{w}), \ where \ \epsilon_\theta(\mathbf{w}) := \mathbb{E}_{\mathbf{x},y \sim \mathbb{P}_\theta} \left[ \text{sign}(\langle \mathbf{w}, \mathbf{x} \rangle) \neq y \right]. \tag{2}$$

**Normalized margin.** We study is whether algorithms that perfectly fit (i.e. interpolate) their training data can learn models with low robust error. Ideally, we would like to give a result on all classifiers that attain training error zero in terms of the 0-1 loss. However, the inherent discontinuity of this loss would make any such statement sensitive to instabilities and pathologies. For instance, if we do not limit the capacity of our models, we can turn any classifier into an interpolating one by adding "special cases" for the training points, yet intuitively this is not the type of interpolation that we would like to study. To avoid such issues, we replace the 0-1 loss with a common continuous surrogate, the normalize margin, and require it to be strictly positive.

**Definition 4** (Normalized margin). *Let* $\gamma > 0$, *we say a classifier* $\mathbf{w} \in \mathbb{R}^d$ *separates the set* $S = \{\mathbf{x}_i, y_i\}_{i=1}^N$ *with* normalized margin $\gamma$ *if for every* $(\boldsymbol{x}, y) \in S$

$$\frac{y_i \langle \mathbf{w}, \mathbf{x}_i \rangle}{\|\mathbf{w}\|} > \gamma \sqrt{\sigma^2 d}.$$

The $\sqrt{\sigma^2 d}$ scaling of $\gamma$ is roughly proportional to $\|\mathbf{x}\|$ under our data model in Equation (1), and keeps the value of $\gamma$ comparable across growing values of $d$.

## 2.2 MAIN RESULT

Equipped with the necessary definitions, we now state and discuss our main result.

**Theorem 1.** *For any sample sizes* $N_1, N_2 > 65$, *margin lower bound* $\gamma \leq \frac{1}{4\sqrt{N}}$, *target robust error* $\epsilon > 0$, *and coefficients* $\theta_1 = 1$, $\theta_2 > -\frac{N_1 \gamma}{\sqrt{288 N_2}}$, *there exist parameters* $r_c, r_s > 0$, $d > N$, *and* $\sigma > 0$ *such that the following holds for the Linear Two Environment Problem (Definition 2) with these parameters.*

1. **Invariance is attainable**. *Algorithm 1 maps* $(S_1, S_2)$ *to a linear classifier* $\mathbf{w}$ *such that with probability at least* $\frac{99}{100}$ *(over the draw $S$), the* robust error *of* $\mathbf{w}$ *is less than* $\epsilon$.

2. **Interpolation is attainable**. *With probability at least* $\frac{99}{100}$, *the estimator* $\mathbf{w}_{\text{mean}} = N^{-1} \sum_{i \in [N]} y_i \mathbf{x}_i$ *separates $S$ with normalized margin (Definition 4) greater than* $\frac{1}{4\sqrt{N}}$.

---

[1]We note that in some settings, more commonly in the fairness literature, $e$ is treated as a feature given to the classifier as input. Our focus is on cases where this is either impossible or undesired. For instance, because at test time $e$ is unobserved or ill-defined (e.g. we obtain data from a new hospital). However, we emphasize that the *leaning rules* we consider have full knowledge of which environment produced each training example

[2]In fact, as we show in Equation (5) in Section 3, learning a model orthogonal to $\boldsymbol{\mu}_s$ is also a necessary condition to minimize the robust error. Thus, attaining guarantees on the robust error also has consequences on invariance of the model, as defined by these criteria. We discuss this further in section F of the appendix.

3. **Interpolation precludes invariance.** *Given $\boldsymbol{\mu}_c$ uniformly distributed on the sphere of radius $r_c$ and $\boldsymbol{\mu}_s$ uniformly distributed on a sphere of radius $r_s$ in the subspace orthogonal to $\boldsymbol{\mu}_c$, let $\mathbf{w}$ be any classifier learned from $(S_1, S_2)$ as per Definition 2. If $\mathbf{w}$ separates $S$ with normalized margin $\gamma$, then with probability at least $\frac{99}{100}$ (over the draw of $\boldsymbol{\mu}_c, \boldsymbol{\mu}_s$, and the sample), the robust error of $\mathbf{w}$ is at least $\frac{1}{2}$.*

Theorem 1 shows that if a learning algorithm for overparameterized linear classifiers always separates its training data, then there exist natural settings for which the algorithm completely fails to learn a robust classifier, and will therefore fail on multiple other invariance and fairness objectives. Furthermore, in the same setting this failure is avoidable, as there exists an algorithm (that *necessarily* does not always separate its training data) which successfully learns an invariant classifier. This result has deep implications for theoreticians attempting to prove finite-sample invariant learning guarantees: it shows that—in the fundamental setting of linear classification—no interpolating algorithm can have guarantees as strong as non-interpolating algorithms such as Algorithm 1.

Importantly, Theorem 1 requires interpolating invariant classifiers to *exist*—and shows that these classifiers are information-theoretically *impossible to learn*. In particular, the first part of the theorem implies that the Bayes optimal invariant classifier $\boldsymbol{w} = \boldsymbol{\mu}_c$ has robust test error at most $\epsilon$. Therefore, for all $\epsilon < \frac{1}{100N}$ we have that $\boldsymbol{\mu}_c$ interpolates $S$ with probability $> \frac{99}{100}$. Furthermore, a short calculation (see Appendix C.1) shows that (for $r_c, r_s, d$ and $\sigma$ satisfying Theorem 1) the normalized margin of $\boldsymbol{\mu}_c$ is $\Omega((N + \sqrt{N_2}/\gamma)^{-\frac{1}{2}})$. However, we prove that—due to the high-dimensional nature of the problem—no algorithm can use $(S_1, S_2)$ to reliably distinguish the invariant interpolator from other interpolators with similar or larger margin. This learnability barrier strongly leverages our random choice of $\boldsymbol{\mu}_c, \boldsymbol{\mu}_s$, without which the (fixed) vector $\boldsymbol{\mu}_c$ would be a valid learning output.

We establish Theorem 1 with three propositions, each corresponding to an enumerated claim in the theorem: (1) Proposition 2 (in §4) establishes that invariance is attainable, (2) Proposition 3 (Appendix C) establishes that interpolation is attainable, and (3) Proposition 1 (in §3) establishes that interpolation precludes invariance. We choose to begin with the latter proposition since it is the main conceptual and technical contribution of our paper. Conversely, Proposition 3 is an easy byproduct of the developments leading up to Proposition 1, and we defer it to the appendix.

With Propositions 1, 2 and 3 in hand, the proof of Theorem 1 simply consists of choosing the free parameters in Theorem 1 ($r_c, r_s, d$ and $\sigma$) based on these propositions such that all the claims in the theorem hold simultaneously. For convenience we take $\sigma^2 = 1/d$. Then (ignoring constant factors) we pick $r_s^2 \propto \frac{1}{N}$ and $r_c^2 \propto r_s^2 / (1 + \frac{\sqrt{N_2}}{N_1 \gamma})$ in order to satisfy requirements in Propositions 1 and 3. Finally, we take $d$ to be sufficiently large so as to satisfy the remaining requirements, resulting in $d \propto \max\left\{ N^2, \frac{N}{\gamma^2 N_1^2 r_c^2}, \frac{(Q^{-1}(\epsilon))^2}{N_{\min} r_c^4}, \frac{1}{N_{\min}^2 r_c^4} \right\}$, where $N_{\min} = \min\{N_1, N_2\}$ and $Q$ is the Gaussian tail function (see Appendix E for the full proof).

We conclude this section with remarks on the range of parameters under which Theorem 1 holds. The impossibility results in Theorem 1 are strongest when $N_2$ is smaller than $N_1^2 \gamma^2$. In particular, when $N_2 \leq N_1^2 \gamma^2 / 288$, our result holds for all $\theta_2 \in [-1, 1]$ and moreover the core and spurious signal strengths $r_c$ and $r_s$ can be chosen to be of the same order. The ratio $N_2/(N_1^2 \gamma^2)$ is small *either* when one group is under-represented (i.e., $N_2 \ll N_1$) *or* when considering large margin classifiers (i.e., $\gamma$ of the order $1/\sqrt{N}$). Moreover, unlike prior work on barriers to robustness (e.g., Sagawa et al., 2020b; Nagarajan et al., 2021), our result continue to hold even for balanced data and arbitrarily low margin, provided $\theta_2$ is close to 0 and the core signal is sufficiently weaker than the spurious signal. Notably, the normalized margin $\gamma$ can be arbitrarily small while the maximum achievable margin is always at least of the order of $\frac{1}{\sqrt{N}}$. Therefore, we believe that Theorem 1 essentially precludes any interpolating learning rule from being consistently invariant.

## 3 INTERPOLATING MODELS CANNOT BE INVARIANT

In this section we prove the third claim in Theorem 1: for essentially any nonzero value of the normalized margin $\gamma$, there are instances of the Linear Two Environment Problem (Definition 2) where with high probability, learning algorithms that return linear classifiers attaining normalized margin at least $\gamma$ must incur a large robust error. The following proposition formalizes the claim; we sketch the proof below and provide a full derivation in Appendix B.3.

**Proposition 1.** *For $\sigma = 1/\sqrt{d}$, $\theta_1 = 1$, there are universal constants $c_r \in (0,1)$ and $C_d, C_r \in (1, \infty)$, such that, for any target normalized $\gamma$, $\theta_2 > -N_1\gamma/\sqrt{288 N_2}$, and failure probability $\delta \in (0,1)$, if*

$$\max\{r_s^2, r_c^2\} \le \frac{c_r}{N} \;\; , \;\; \frac{r_s^2}{r_c^2} \ge C_r \left(1 + \frac{\sqrt{N_2}}{N_1\gamma}\right) \quad and \tag{3}$$

$$d \ge C_d \frac{N}{\gamma^2 N_1^2 r_c^2} \log \frac{1}{\delta}, \tag{4}$$

*then with probability at least $1-\delta$ over the drawing of $\boldsymbol{\mu}_c$, $\boldsymbol{\mu}_s$ and $(S_1, S_2)$ as described in Theorem 1, any $\hat{\mathbf{w}} \in \mathbb{R}^d$ that is a measurable function of $(S_1, S_2)$ and separates the data with normalized margin larger than $\gamma$ has robust error at least $0.5$.*

*Proof sketch.* We begin by noting that for any fixed $\theta$, the error of a linear classifier $\mathbf{w}$ is

$$\epsilon_\theta(\mathbf{w}) = Q\left(\frac{\langle \mathbf{w}, \boldsymbol{\mu}_c \rangle + \theta \langle \mathbf{w}, \boldsymbol{\mu}_s \rangle}{\sigma \|\mathbf{w}\|}\right) = Q\left(\frac{\langle \mathbf{w}, \boldsymbol{\mu}_c \rangle}{\sigma \|\mathbf{w}\|}\left(1 + \theta \frac{\langle \mathbf{w}, \boldsymbol{\mu}_s \rangle}{\langle \mathbf{w}, \boldsymbol{\mu}_c \rangle}\right)\right), \tag{5}$$

where $Q(t) := \mathbb{P}(\mathcal{N}(0;1) > t)$ is the Gaussian tail function. Consequently, when $\langle \mathbf{w}, \boldsymbol{\mu}_s \rangle / \langle \mathbf{w}, \boldsymbol{\mu}_c \rangle \ge 1$ it is easy to see that $\epsilon_\theta(\mathbf{w}) = 1/2$ for some $\theta \in [-1,1]$ and therefore the robust error is at least $\frac{1}{2}$; we prove that $\langle \mathbf{w}, \boldsymbol{\mu}_s \rangle / \langle \mathbf{w}, \boldsymbol{\mu}_c \rangle \ge 1$ indeed holds with high probability under the proposition's assumptions. Our proof has two key parts: (a) restricting the set of classifiers to the linear span of the data and (b) lower bounding the minimum value of $\langle \mathbf{w}, \boldsymbol{\mu}_s \rangle / \langle \mathbf{w}, \boldsymbol{\mu}_c \rangle$ for classifier in that linear span.

For the first part of the proof we use the spherical distribution of $\boldsymbol{\mu}_c$ and $\boldsymbol{\mu}_s$ and concentration of measure to show that (with high probability) any component of $\mathbf{w}$ chosen outside the linear span of $\{\mathbf{x}_i\}_{i \in [N]}$ will have negligible effect on the predictions of the classifier. To explain this fact, let $\boldsymbol{P}_\perp$ denote the projection operator to the orthogonal complement of the data, so that $\boldsymbol{P}_\perp \mathbf{w}$ is the component of $\mathbf{w}$ orthogonal to the data and $\langle \boldsymbol{P}_\perp \mathbf{w}, \boldsymbol{\mu}_c \rangle = \left\langle \mathbf{w}, \frac{\boldsymbol{P}_\perp \boldsymbol{\mu}_c}{\|\boldsymbol{P}_\perp \boldsymbol{\mu}_c\|} \right\rangle \|\boldsymbol{P}_\perp \boldsymbol{\mu}_c\|$. Conditional on $(S_1, S_2)$ and the learning rule's random seed, the vector $\boldsymbol{P}_\perp \boldsymbol{\mu}_c / \|\boldsymbol{P}_\perp \boldsymbol{\mu}_c\|$ is uniformly distributed on a unit sphere of dimension $d - N$ while the vector $\mathbf{w}$ is deterministic. Assuming without loss of generality that $\|\mathbf{w}\| = 1$, concentration of measure on the sphere implies that $|\langle \mathbf{w}, \frac{\boldsymbol{P}_\perp \boldsymbol{\mu}_c}{\|\boldsymbol{P}_\perp \boldsymbol{\mu}_c\|} \rangle|$ is (with high probability) bounded by roughly $1/\sqrt{d}$, and therefore $|\langle \boldsymbol{P}_\perp \mathbf{w}, \boldsymbol{\mu}_c \rangle|$ is roughly of the order $r_c/\sqrt{d}$. For sufficiently large $d$ (as required by the proposition), this inner product would be negligible, meaning that $\langle \mathbf{w}, \boldsymbol{\mu}_c \rangle$ is roughly the same as $\langle (I - \boldsymbol{P}_\perp)\mathbf{w}, \boldsymbol{\mu}_c \rangle$, and $(I - \boldsymbol{P}_\perp)\mathbf{w}$ is in the span of the data. The same argument applies to $\boldsymbol{\mu}_s$ as well.

In the second part of the proof, we consider classifiers of the form $\mathbf{w} = \sum_{i \in [N]} \beta_i y_i \mathbf{x}_i$ (which parameterizes the linear span of the data) and minimize $\langle \mathbf{w}, \boldsymbol{\mu}_s \rangle / \langle \mathbf{w}, \boldsymbol{\mu}_c \rangle$ over $\beta \in \mathbb{R}^N$ subject to the constraint that $\mathbf{w}$ has normalize margin of at least $\gamma$. To do so, we first use concentration of measure to argue that it is sufficient to lower bound $\sum_{i \in [N_1]} \beta_i$ subject to the margin constraint and $\|\mathbf{w}\|^2 \le 1$, which is convex in $\beta$—we obtain this lower bound by analyzing the Lagrange dual of the problem of minimizing $\sum_{i \in [N_1]} \beta_i$ subject to these constraints.

Overall, we show a high-probability lower bound on $\frac{\langle \mathbf{w}, \boldsymbol{\mu}_s \rangle}{\langle \mathbf{w}, \boldsymbol{\mu}_c \rangle}$ that (for sufficiently high dimensions) scales roughly as $\frac{r_s^2 N_1 \gamma}{r_c^2 \sqrt{N_2}}$. For parameters satisfying Equation (3) we thus obtain $\frac{\langle \mathbf{w}, \boldsymbol{\mu}_s \rangle}{\langle \mathbf{w}, \boldsymbol{\mu}_c \rangle} \ge 1$, completing the proof. $\qquad\square$

**Implication for invariance-inducing algorithms.** Our proof implies that any interpolating algorithm should fail at learning invariant classifiers. This alone does not necessarily imply that specific algorithms proposed in the literature for learning invariant classifiers fail, as they may not be interpolating. Yet our simulations in Section 5 show that several popular algorithms proposed for eliminating spurious features are indeed interpolating in the overparameterized regime. We also give a formal statement in Appendix G regarding the IRMv1 penalty (Arjovsky et al., 2019), showing that it is biased toward large margins when applied to separable datasets. Our results may seem discouraging for the development of invariance-inducing techniques using overparameterized models. It is natural to ask what type of methods *can* provably learn such models, which is the topic of the next section.

---

**Algorithm 1** Two Phase Learning of Overparameterized Invariant Classifiers

---

**Input:** Datasets $(S_1, S_2)$ and an invariance constraint function family $\mathcal{F}(\cdot, \cdot)$

**Output:** A classifier $f_{\mathbf{v}}(\mathbf{x})$

    Draw subsets of data without replacement $S_e^{\text{train}} \subset S_e$ for $e \in \{1, 2\}$ where $\left| S_e^{\text{train}} \right| = N_e/2$

    Stage 1: Calculate $\mathbf{w}_e = 2N_e^{-1} \sum_{(\mathbf{x},y) \in S_e^{\text{train}}} y\mathbf{x}$ for each $e \in \{1, 2\}$

    Define $S_e^{\text{fine}} = S_e \setminus S_e^{\text{trn}}$ for $e \in \{1, 2\}$ and $S^{\text{post}} = S_1^{\text{fine}} \cup S_2^{\text{fine}}$

    Stage 2: Return $f_{\mathbf{v}}(\mathbf{x}) = \langle v_1 \cdot \mathbf{w}_1 + v_2 \cdot \mathbf{w}_2, \mathbf{x} \rangle$ that solves

$$\text{maximize} \sum_{(\mathbf{x},y) \in S^{\text{post}}} y f_{\mathbf{v}}(\mathbf{x}) \quad \text{subject to} \quad \|\mathbf{v}\|_\infty = 1 \quad \text{and} \quad f_{\mathbf{v}} \in \mathcal{F}(S_1^{\text{fine}}, S_2^{\text{fine}})$$

---

## 4   A Provably Invariant Overparameterized Estimator

We now turn to propose and analyze an algorithm (Algorithm 1) that provably learns an overparametrized linear model with good robust accuracy in our setup. Our approach is a two-staged learning procedure that is conceptually similar to some recently proposed methods (Rosenfeld et al., 2022; Veldanda et al., 2022; Kirichenko et al., 2022; Menon et al., 2021; Kumar et al., 2022; Zhang et al., 2022). In Section 5 we validate our algorithm on simulations and on the Waterbirds dataset Sagawa et al. (2020a), but we leave a thorough empirical evaluation of the techniques described here to future work.

Let us describe the operation of Algorithm 1. First, we evenly[3] split the data from each environment into the sets $S_e^{\text{train}}, S_e^{\text{post}}$, for $e \in \{1, 2\}$. The two stages of the algorithm operate on different splits of the data as follows.

1. **"Training" stage**: We use $\{S_e^{\text{train}}\}$ to fit overparameterized, interpolating classifiers $\{\mathbf{w}_e\}$ *separately* for each environment $e \in \{1, 2\}$.

2. **"Post-processing" stage**: We use the second portion of the data $\left(S_1^{\text{post}}, S_2^{\text{post}}\right)$ to learn an invariant linear classifier over a new representation, which concatenates the outputs of the classifiers in the first stage. In particular, we learn this classifier by maximizing a score (i.e., minimizing an empirical loss), subject to an empirical version of an invariance constraint. For generality we denote this constraint by membership in some set of functions $\mathcal{F}(S_1^{\text{post}}, S_2^{\text{post}})$.

Crucially, the invariance penalty is only used in the second stage, in which we are no longer in the overparamterized regime since we are only fitting a two-dimensional classifier. In this way we overcome the negative result from Section 3.

While our approach is general and can handle a variety of invariance notions (we discuss some of them in Appendix F), we analyze the algorithm under the Equal Opportunity (EOpp) criterion (Hardt et al., 2016). Namely, for a model $f : \mathbb{R}^d \to \mathbb{R}$ we write:

$$\mathcal{F}(S_1^{\text{fine}}, S_2^{\text{fine}}) = \left\{ f : \hat{T}_1(f) = \hat{T}_2(f) \right\}, \quad \text{where } \hat{T}_e(f) := \frac{4}{N_e} \sum_{(\mathbf{x},y) \in S_e^{\text{fine}}: y=1} f(\mathbf{x}).$$

This is the empirical version of the constraint $\mathbb{E}_{\mathbb{P}_{\theta_1}}[f(x)|y=1] = \mathbb{E}_{\mathbb{P}_{\theta_2}}[f(x)|y=1]$. From a fairness perspective (e.g., thinking of a loan application), this constraint ensures that the "qualified" members (i.e., those with $y = 1$) of each group receive similar predictions, on average over the entire group.

We now turn to providing conditions under which Algorithm 1 successfully learns an invariant predictor. The full proof for the following proposition can be found in section D.1 of the appendix. While we do not consider the following proposition very surprising, the fact that it gives a finite sample learning guarantee means it does not directly follow from existing work (discussed in §6 below) that mostly assume inifinite sample size.

---

[3] The even split is used here for simplicity of exposition, and our full proof does not assume it. In practice, allocating more data to the first-stage split would likely perform better.

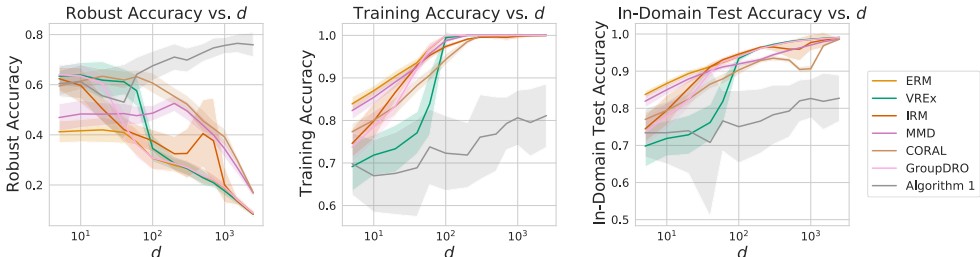

Figure 1: Numerical validation of our theoretical claims. Invariance inducing methods improve robust accuracy compared to ERM in low values of $d$, but their ability to do so is diminished as $d$ grows (top plot) and they enter the interpolation regime, as seen on the bottom plot for $d > 10^2$. Algorithm 1 learns robust predictors as $d$ grows and does not interpolate.

**Proposition 2.** *Consider the Linear Two Environment Problem (Definition 2), and further suppose that $|\theta_1 - \theta_2| > 0.1$.[4] There exist universal constants $C_p, C_c, C_s \in (1, \infty)$ such that the following holds for every target robust error $\epsilon > 0$ and failure probability $\delta \in (0, 1)$. If $N_{\min} := \min\{N_1, N_2\} \geq C_p \log(4/\delta)$ for some $C_p \in (1, \infty)$,[5]*

$$r_s^2 \geq C_s \sqrt{\log \frac{68}{\delta}} \frac{\sigma^2 \sqrt{d}}{N_{\min}}, \ r_c^2 \geq C_c \sigma^2 \sqrt{\log \frac{68}{\delta}} \max \left\{ Q^{-1}(\epsilon) \sqrt{\frac{d}{N_{\min}}}, \frac{\sqrt{d}}{N_{\min}}, \frac{r_s^2}{N_{\min} r_c^2} \right\}, \quad (6)$$

$$\text{and } d \geq \log \frac{68}{\delta} \tag{7}$$

*then, with probability at least $1 - \delta$ over the draw of the training data and the split of the data between the two stages of learning, the robust error of the model returned by Algorithm 1 does not exceed $\epsilon$.*

*Proof sketch.* Writing down the error of $f_{\mathbf{v}} = v_1 \cdot \mathbf{w}_1 + v_2 \cdot \mathbf{w}_2$ under $\mathbb{P}_\theta$, it can be shown that to obtain the desired bound on the robust error of the classifier returned by Algorithm 1, we must upper bound the ratio

$$\frac{(v_1^\star \theta_1 + v_2^\star \theta_2)\|\boldsymbol{\mu}_s\|^2 + \langle \boldsymbol{\mu}_s, v_1^\star \bar{\mathbf{n}}_1 + v_2^\star \bar{\mathbf{n}}_2 \rangle}{(v_1^\star + v_2^\star)\|\boldsymbol{\mu}_c\|^2 + \langle \boldsymbol{\mu}_c, v_1^\star \bar{\mathbf{n}}_1 + v_2^\star \bar{\mathbf{n}}_2 \rangle},$$

when $\bar{\mathbf{n}}_e$ is the mean of Gaussian noise vectors, and $v_1^\star$ and $v_2^\star$ are the solutions to the optimization problem in Stage 2 of Algorithm 1. The terms involving inner-products with the noise terms are zero-mean and can be bounded using standard Gaussian concentration arguments. Therefore, the main effort of the proof is upper bounding

$$\frac{v_1^\star \theta_1 + v_2^\star \theta_2}{v_1^\star + v_2^\star} \cdot \frac{\|\boldsymbol{\mu}_s\|^2}{\|\boldsymbol{\mu}_c\|^2}.$$

To this end, we leverage the EOpp constraint. The population version of this constraint (corresponding to infinite $N_1$ and $N_2$) implies that $v_1^\star \theta_1 + v_2^\star \theta_2 = 0$. For finite sample sizes, we use standard Gaussian concentration and the Hanson-Wright inequality to show that the empirical EOpp constraint implies that $|v_1^\star \theta_1 + v_2^\star \theta_2|$ goes to zero as the sample sizes increase. Furthermore, we argue that $|v_1^\star + v_2^\star| \geq |\theta_1 - \theta_2|/2$, implying that—for appropriately large sample sizes—the above ratio indeed goes to zero. □

## 5 EMPIRICAL VALIDATION

The empirical observations that motivated this work can be found across the literature. We therefore focus our simulations on validating the theoretical results in our simplified model. We also evaluate Algorithm 1 on the Waterbirds dataset, where the goal is not to show state-of-the-art results, but rather to observe whether our claims hold beyond the Linear Two Environment Problem.

---

[4]Intuitively, if $|\theta_1 - \theta_2| = 0$ then the two training environments are indistinguishable and we cannot hope to identify that the correlation induced by $\boldsymbol{\mu}_s$ is spurious. Otherwise, we expect $|\theta_1 - \theta_2|$ to have a quantifiable effect on our ability to generalize robustly. For simplicity of this exposition we assume that the gap is bounded away from zero; the full result in the Appendix is stated in terms of $|\theta_1 - \theta_2|$.

[5]This assumption makes sure we have some positive labels in each environment.

## 5.1 SIMLUATIONS

**Setup.** We generate data as described in Theorem 1 with two environments where $\theta_1 = 1, \theta_2 = 0$ (see Figure 4 in the appendix for results of the same simulation when $\theta = -\frac{1}{2}$). We further fix $r_c = 1$ and $r_c = 2$, while $N_1 = 800$ and $N_2 = 100$. We then take growing values of $d$, while adjusting $\sigma$ so that $(r_c/\sigma)^2 \propto \sqrt{d/N}$.[6] For each value of $d$ we train linear models with IRMv1 (Arjovsky et al., 2019), VREx (Krueger et al., 2021), MMD (Li et al., 2018), CORAL (Sun & Saenko, 2016), GroupDRO (Sagawa et al., 2020a), implemented in the Domainbed package (Gulrajani & Lopez-Paz, 2021). We also train a classifier with the logistic loss to minimize empirical error (ERM), and apply Algorithm 1 where the "post-processing" stage trains a linear model over the two-dimensional representation using the VREx penalty to induce invariance. We repeat this for 15 random seeds for drawing $\boldsymbol{\mu}_c, \boldsymbol{\mu}_s$ and the training set.

**Evaluation and results.** We compare the robust accuracy and the train set accuracy of the learned classifiers as $d$ grows. First, we observe that all methods except for Algorithm 1 attain perfect accuracy for large enough $d$, i.e., they interpolate. We further note that while invariance-inducing methods give a desirable effect in low dimensions (the non-interpolating regime)—significantly improving the robust error over ERM—they become aligned with ERM in terms of robust accuracy as they go deeper into the interpolation regime (indeed, IRM essentially coincides with ERM for larger $d$). This is an expected outcome considering our findings in section 3, as we set here $N_1$ to be considerably larger than $N_2$.

## 5.2 WATERBIRDS DATASET

We evaluate Algorithm 1 on the Waterbirds dataset (Sagawa et al., 2020a), which has been previously used to evaluate the fairness and robustness of deep learning models.

**Setup.** Waterbirds is a synthetically created dataset containing images of water- and land-birds overlaid on water and land background. Most of the waterbirds (landbirds) appear in water (land) backgrounds, with a smaller minority of waterbirds (landbirds) appearing on land (water) backgrounds. We set up the problem following previous work (Sagawa et al., 2020b; Veldanda et al., 2022), where a logistic regression model is trained over random features extract from a fixed pretrained ResNet-18. Please see Appendix H for details.

**Fairness.** We use the image background type (water or land) as the sensitive feature, denoted $A$, and consider the fairness desiderata of Equal Opportunity Hardt et al. (2016), i.e., the false negative rate (FNR) should be similar for both groups. Towards this, we use the MinDiff penalty term (Prost et al., 2019). The

**Evaluation.** We compare the following methods: **(1) Baseline**: Learning a linear classifier $\mathbf{w}$ by minimizing $\mathcal{L}_p + \lambda \cdot \mathcal{L}_M$, where $\mathcal{L}_p$ is the standard binary cross entropy loss and $\mathcal{L}_M$ is the MinDiff penalty; **(2) Algorithm 1**: In the first stage, we learn group-specific linear classifiers $\mathbf{w}_0, \mathbf{w}_1$ by minimizing $\mathcal{L}_p$ on the examples from $A = 0$ and $A = 1$, respectively. In the second stage we learn $\boldsymbol{v} \in \mathbb{R}^2$ by minimizing $\mathcal{L}_p + \lambda \cdot \mathcal{L}_M$ on examples the entire dataset, where the new representation of the data is $\tilde{X} = [\langle w_1, X \rangle, \langle w_2, X \rangle] \in \mathbb{R}^2$.[7]

**Results.** Our main objective is to understand the effect of the fairness penalty. Toward this, for each method we compare both the test error and the test FNR gap when using either $\lambda = 0$ (no regularization) or $\lambda = 5$. The results are summarized in Figure 2. We can see that for the baseline approach, the fairness penalty successfully reduces the FNR gap when the classifier is not interpolating. However, as our negative result predicts and as previously reported in Veldanda et al. (2022), the fairness penalty becomes ineffective in the interpolating regime ($d \geq 1000$). On the other hand, for our two-phased algorithm, the addition of the fairness penalty reduces does reduce the FNR gap with an average relative improvement of 20%; crucially, this improvement is independent of $d$.

---

[6]This is to keep our parameters within the regime where benign overfitting occurs.

[7]This is basically Algorithm 1 with the following minor modifications: (1) The $\mathbf{w}_e$'s are computed via ERM, rather than simply taken to be the mean estimators; (2) Since the FNR gap penalty is already computed w.r.t. a small number of samples, we avoid splitting the data and use the entire training set for both phases; (3) we convert the constrained optimization problem into an unconstrained problem with a penalty term.

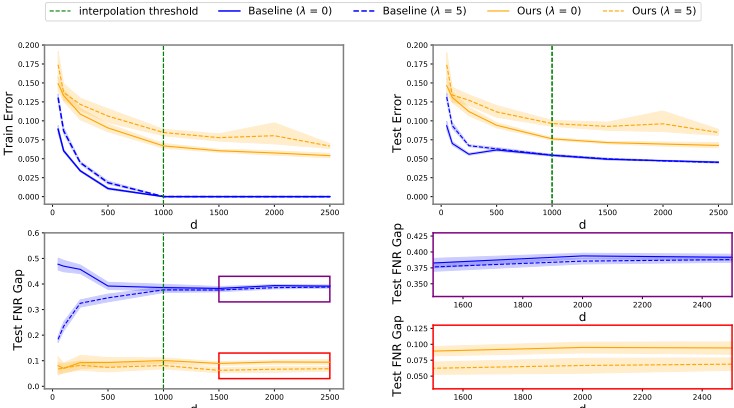

Figure 2: Results for the Waterbirds dataset (Sagawa et al., 2020a). **Top row**: Train error (left) and test error (right). The train error is used to identify the interpolation threshold for the baseline method (approximately $d = 1000$). **Bottom row**: Comparing the FNR gap on the test set (left), with zoomed-in versions on the right.

## 6 DISCUSSION AND ADDITIONAL RELATED WORK

In terms of formal results, most existing guarantees about invariant learning algorithms rely on the assumption that infinite training data is available (Arjovsky et al., 2019; Wald et al., 2021; Veitch et al., 2021; Puli et al., 2021; Rosenfeld et al., 2021; Diskin et al., 2021). Wang et al. (2022); Chen et al. (2022) analyze algorithms that bear resemblance to Algorithm 1 as they first project the data to a lower dimension and then fit a classifier. While these algorithms deal with more general assumptions in terms of the number of environments, number of spurious features, and noise distribution, the fact that their guarantees assume infinite data prevents them from being directly applicable to Algorithm 1. A few works with results on finite data are Ahuja et al. (2021); Parulekar et al. (2022) (and also Efroni et al. (2022) who work on related problems in the context of sequential decision making) that characterize the sample complexity of methods that learn invariant classifiers. However, they do not analyze the overparameterized cases we are concerned with.

Negative results about learning overparameterized robust classifiers have been shown for methods based on importance weighting (Zhai et al., 2022) and max-margin classifiers (Sagawa et al., 2020b). Our result is more general, applying to any learning algorithm that separates the data with arbitrarily small margins, instead of focusing on max-margin classifiers or specific algorithms. While we focus on the linear case, we believe it is instructive, as any reasonable method is expected to succeed in that case. Nonetheless, we believe our results can be extended to non-linear classifiers, and we leave this to future work.

One take-away from our result is that while low training loss is generally desirable, overfitting to the point of interpolation can significantly hinder invariance-inducing objectives. This means one cannot assume a typical deep learning model with an added invariance penalty will indeed achieve any form of invariance; this fact also motivates using held-out data for imposing invariance, as in our Algorithm 1 as well as several other two-stage approaches mentioned above.

Our work focuses theory underlying a wide array of algorithms, and there are natural follow-up topics to explore. One is to conduct a comprehensive empirical comparison of two-stage methods along with other methods that avoid interpolation, e.g., by subsampling data (Idrissi et al., 2022; Chatterji et al., 2022). Another interesting topic is whether there are other model properties that are incompatible with interpolation. For instance, recent work (Carrell et al., 2022) connects the generalization gap and calibration error on the training distribution. We also note that our focus in this paper was not on types of invariance that are satisfiable by using clever data augmentation techniques (e.g. invariance to image translation), or the design of special architectures (e.g. Cohen & Welling (2016); Lee et al. (2019); Maron et al. (2019)). These methods carefully incorporate a-priori known invariances, and their empirical success when applied to large models may suggest that there are lessons to be learned for the type of invariant learning considered in our paper. These connections seem like an exciting avenue for future research.

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
