# OpenReview forum: "Malign Overfitting: Interpolation and Invariance are Fundamentally at Odds"
_ICLR.cc/2023/Conference — ICLR 2023 poster_

### Official Review · Reviewer_VPKR · 2022-10-24

**Confidence:** 5
**Correctness:** 4
**Technical Novelty And Significance:** 3
**Empirical Novelty And Significance:** 2
**Recommendation:** 8

**Clarity, Quality, Novelty And Reproducibility:**

Quality: High
Clarity: General writing is great, the proof technique and specific parameterizations in the first theorem could be made much clearer.
Novelty: First half - excellent, Second half - already known
Reproducibility: Good

**Strength And Weaknesses:**

**I like this paper, and I think it is well-written. There are some non-insignificant weaknesses, but overall I am in favor of acceptance. But, there are some changes/missing discussion I would like to see.**

The primary theorem is asking a really interesting question, and I think the result, while not too surprising, is valuable. Actually, what I find more surprising is the fact that the data is linearly separable at all with the chosen parameters (high dimensions are weird! I had to simulate this to convince myself that it was true). Generally, and along the same lines, I think this work would significantly benefit from a high-level description of the proof and giving an overview of the general order of the parameters that are necessary to make the proof work. Seeing "there exist parameters a, b, c" without a sense of scale feels extremely unsatisfying and sometimes gives me the impression that the result may not be as interesting **(e.g., we could take $\mu_c \to 0$ and the result would be trivial, no?).** It also seems unusual to have the means be separated by a term which varies with N but is constant as a function of the dimension... I would appreciate an indication of how robust this result is to different ranges of the parameters, even if just an intuition. For example, what if we choose $\theta_2 \neq 0$? How small would it have to be for the result to still hold? What if $\theta_1 \in o(1)$?

Unfortunately I think the second part of the paper is quite a bit weaker. First of all, the model studied here is a specific case of the model from Rosenfeld et al. where the spurious means are collinear. This is pretty unrealistic (I admit this is a bit subjective), but it also immediately suggests a trivial method for invariance: just do PCA on the features to identify the shared spurious subspace, then project it out. However, **this approach has already been suggested and studied for a more general model by two works [1, 2], which both show invariance with only two environments.** So the second half of the paper unfortunately is giving a weaker result than one that is already known. The proposed two-stage algorithm is a reasonable idea, and I appreciate the authors' efforts to acknowledge existing work which does similar things---but it would be helpful to specifically explain how this approach differs rather than just giving a long list of citations.

In summary:

* The first half of the paper asks an interesting question and gives a very nice result, though it requires a pretty specific parameterization which I think needs to be clarified for an updated draft.
* The second half of the paper is already known, though the slightly different algorithm and experiments do provide some value showing that this approach is reasonable.

On the basis of the first half of this work alone I think this paper should be accepted, and I think this work would be much stronger if the results there could be made more general or at least clearer.

**One last note: Given the restricted setting of the theory, I think the title would be much more accurately stated as a possibility, such as "can/may be/is sometimes fundamentally at odds". As it is written it feels heavily overclaimed.**

[1] Provable Domain Generalization via Invariant-Feature Subspace Recovery. Wang et al. '22

[2] Iterative Feature Matching: Toward Provable Domain Generalization with Logarithmic Environments. Chen et al. '21


**Summary Of The Paper:**

The paper studies a setting with two environments from a specific instance of the model from Rosenfeld et al. and show that overfitting (i.e., interpolation of the training data) and invariance to the spurious features are fundamentally at odds, and that while both are achievable, they cannot be done simultaneously. They then propose an algorithm which learns to make invariant predictions based on these two environments.

**Summary Of The Review:**

See above

---

> ### Author Response · Authors · 2022-11-11
> **Author Response**
>
> We thank the reviewer for the positive review and highly insightful comments and suggestions. We would like to respond to some of the points raised by the reviewer and note some changes we will make in the manuscript following the reviewer’s comments.
>
> **High level description of the proof.** We have revised the proof sketch of Proposition 1 to incorporate more details, give a better sense of its required steps and the role of the inequalities in the proof statement. Specifically, we the revision will include the term $\frac{r_s^2 N_1 \gamma}{r_c^2 \sqrt{N_2}}$ which is the lower bound we derive for the ratio $<w, \mu_s> / <w, \mu_c>$, and can be plugged in directly to the robust error to obtain meaningful bounds.
>
> **Ranges of parameters where the result holds.** We hope that the above adjustments will portray a better picture of the parameters $r_c, r_s, N_1, N_2, \gamma$ under which the statement holds, reflecting that this is not an ``edge case” with a specially crafted set of parameters.
>
> **Value of $\theta_2$.**  Our current proof uses $\theta_2=0$ mainly for mathematical convenience and other values of $\theta_2\ge -1$ can be accommodated under sufficiently small $N_2 / (N_1^2 \gamma)$. Our revised manuscript will show the result for different values of $\theta_2$.
>
> **Size of $\mu_c$.** Taking $\mu_c \rightarrow 0$ would not align with our statement of Theorem 1. This is because the first part of the theorem requires that for any $\epsilon > 0$ we will be able to achieve robust error at most $\epsilon$, while the Bayes-optimal robust error (attained by $w=\mu_c$) will go to $1/2$ as $\mu_c\to 0$. Hence a vanishing $\mu_c$ would trivially guarantee high robust error, but it won’t be able to accommodate low robust error for some non-interpolating learning rule.
>
> **Vanishing $\theta_1$.** We are not sure we understood what the reviewer meant by $\theta_1=o(1)$: if we choose $\theta_1$ to be very small while also keeping $\theta_2=0$ then the spurious feature will have vanishing presence in the training data, making learning an invariant predictor trivial.
>
> **Distinction between result on Alg. 1 and previous works.** While we agree that algorithms based on a two-step “project and fit” type of procedure have been analyzed, to the best of our knowledge their guarantees hold under infinite data (and particularly not in the overparameterized regime). Hence we could not see how one may plug in existing results to prove the claims in Proposition 2. We also note that proving the robustness of Alg. 1 under infinite data is quite straightforward, and most of our work in this part of the paper goes toward deriving a finite sample bound. We also looked into adapting the results on finite sample bounds for invariant learning from Ahuja et. al. and Parulekar et. al. (references in the discussion of the paper) to the overparameterized case, yet could not come up with a simpler proof than the one provided for Alg. 1. With that being said, we would be happy to take a suggestion on applying existing results to simplify this part of the paper. We thank the reviewer for pointing out relevant work that we regretfully left out of the references; we will add them in the revision, emphasizing that the contribution here is in the finite sample bound.
>
> **What we mean by “fundamentally at odds”.** It is certainly true that our results are valid only for a particular linear classification setting. However, to our knowledge this is the simplest model that one can write down for learning under a spurious correlation. In our opinion, any invariant learning algorithm that does not succeed in this (solvable) fundamental model, has a fundamental flaw - even if it happens to be successful at other, more complex settings. Therefore, since we prove that any interpolating algorithm will fail in our setting, we believe that the paper’s titular claim is justified. That said, we acknowledge that the word “fundamental” could mean different things to different people - if the reviewer still feels the title over-claims, we are open to modifying it in the revision.

---

> > ### Comment · Reviewer_VPKR · 2022-11-15
> > **Update**
> >
> > Thanks, I look forward to reading the updated (more general) proof outlines (I don't see an updated version currently, is it yet to be uploaded or is it not available until the end of open discussion?)
> >
> > I think you're correct that the prior work only considers the limit of infinite samples per environment, but unless I'm mistaken it would also be straightforward to extend these results to the finite sample case by bounding the error in recovery of the principal components with Davis-Kahan. So while this work may be the first to explicitly write out a finite-sample bound, I still think the same idea exists in prior work. I haven't worked out what the exact rates would be, so it's hard to directly compare them. My other concern was the fact that this is specific to two environments where the varying means are co-linear and the covariances are spherical, which feels quite restrictive---whereas prior work is for arbitrary means and covariances.
> >
> > That being said, this *is* the first work I'm aware of to write out a finite-sample result, and it is a different approach which could be useful for future algorithmic development. So I'm satisfied so long as the updated manuscript properly contextualizes this result with prior work.
> >
> > **"In our opinion, any invariant learning algorithm that does not succeed in this (solvable) fundamental model, has a fundamental flaw - even if it happens to be successful at other, more complex settings."**
> >
> > I don't disagree with this. But the argument that something fails in a simple setting, while a very strong statement, doesn't mean that it fails in *all* (or even *most*) settings. And I think to claim that two things are "fundamentally at odds" implies that they cannot (or usually cannot) co-occur, or that there is always a trade-off. But this is very clearly not the case in many settings of interest. So while I think that the simplicity of the learning task is a strength---rather than a weakness---for your theoretical results, I still don't think the current title is warranted.
> >
> > Furthermore, *even in this simple model, they **can** co-occur*. As you yourself pointed out in your rebuttal to another reviewer, it is not the case that there does not *exist* an invariant and interpolating classifier, but rather that it cannot be learned with high probability. So really, your results are showing that "learning an invariant classifier" and "learning an interpolating classifier" are at odds, and only under this specific model with a particular parameterization.

---

> > > ### Author Response · Authors · 2022-11-16
> > > **Thank You for the Continued Engagement, Revision has Been Uploaded**
> > >
> > > Thank you for the continued engagement and further useful comments. We have uploaded a revision now, and hope you will find it properly addresses your suggestions.
> > > Specifically, we include a more detailed general proof sketch for Proposition 1 (our main result), treatment of values $\theta_2 \neq 0$ and a discussion (a bit short due to space constraints) of previous work that focused on environment complexity of more general settings. The key edits are marked in blue for convenience. An additional point that may be of interest is Appendix G, where we analyze the IRMv1 penalty in linear models on separable data.
> > >
> > > Finally, we have changed the title of the paper to “Malign Overfitting: Interpolation Can Provably Preclude Invariance,” and made appropriate changes in other places. We think that this reflects the results of the paper in a more concrete manner. Nonetheless, we still believe that our result constitutes a fundamental separation between interpolating and non-interpolating invariant classifiers when viewed from the perspective of designing algorithms with learning guarantees. In particular, our result means that - for the fundamental setting of linear classification - no method can learn interpolating invariant classifiers with sample complexity guarantees as strong as those enjoyed by methods learning non-interpolating invariant classifiers.

---

### Official Review · Reviewer_p2tP · 2022-10-25

**Confidence:** 4
**Correctness:** 4
**Technical Novelty And Significance:** 2
**Empirical Novelty And Significance:** 2
**Recommendation:** 5

**Clarity, Quality, Novelty And Reproducibility:**

The presentation of the paper is clear. The results are original and reproducible.

**Strength And Weaknesses:**

Strengths:

This paper points out an interesting trade-off between achieving interpolation and invariance.

The authors back up their theoretical claims with convincing simulations.

The paper is well-written and has rigorous proof.

Weaknesses:

Theorem 1 only proves the existence of certain particular values of $r_c,r_s, d, \sigma, \theta_1,\theta_2$. This makes the result relatively weak. In addition, although the idea introduced in the paper is interesting, the constructed learning problem seems "unfair" for standard learning methods. For example, if $\theta_1,\theta_2$ are both positive, then when faced with this particular linear two environment problem, any reasonable learning algorithm without additional prior knowledge should try to learn both $\mu_c$ and $\mu_s$ -- given only the training data set generated based on positive $\theta_1,\theta_2$, who would possibly know that in a new test task the corresponding $\theta$ could be negative?

On the other hand, it is fairly easy to construct algorithms such as Algorithm 1 by utilizing "hidden knowledge" that:
1. In a new test task the corresponding $\theta$ could be negative;
2. $\mu_c$ is "robust" while $\mu_s$ is not;
3. $r_c$ and $r_s$ are different.

Therefore, the significance of the result is questionable.



**Summary Of The Paper:**

This paper demonstrates that interpolation and invariance are fundamentally at odds in overparameterized learning via constructing a linear two environment problem. Specifically, the authors demonstrated that in the constructed learning problem, any interpolating linear model with a positive margin cannot robustly have good out-of-distribution generalization. In contrast, the authors can construct an algorithm (to a certain extent, specifically designed according to the nature of the constructed problem) that is guaranteed to produce robust linear classifiers.



**Summary Of The Review:**

By constructing an example problem, this paper introduces an interesting observation about interpolation and invariance in overparameterized linear classification. However, the significance of the result and the impact of the constructed example problem need a further demonstration.

---

> ### Author Response · Authors · 2022-11-11
> **Author Response**
>
> We thank the reviewer for the valuable comments and time put into reading our work. We also appreciate that the reviewer thinks our results are original. To address the weaknesses claimed by the reviewer we would like to raise the following points:
>
> 1. Please note that our proof does not hold for specific values of $r_c, r_s, d, \sigma$, but a wide range of them. Moreover, the simulations are performed with rather mild underrepresentation ($N2 / N1$) and spuriousness ($r_s / r_c$) parameters. Importantly, the choice of $\theta_1$ and $\theta_2$ is mainly a mathematical convenience; the proof extends to general values, for sufficiently small values of $N_2 / (N_1^2 \gamma)$. These values include $\theta_2 < 0$, for which it may be more intuitive that a reasonable learning algorithm, for instance one seeking best worst-group accuracy, should discard $\mu_s$. We will include a proof supporting a general value of $\theta_2$ in the revised manuscript.
>
> 2. We beg to differ on the reviewer’s view that when $\theta_1, \theta_2$ are positive then the setting is not reasonable. The reviewer is correct in saying that some additional prior knowledge is required, but this prior knowledge is exactly the source of the invariance constraint that algorithms impose. More concretely, we urge the reviewer to consider works such as Invariant Causal Prediction [1], Invariant Risk Minimization [2] and hundreds of subsequent works (including the examples in [3] that exactly suggest the study of linear settings where the spurious feature is in some sense more predictive than the invariant one in both environments), that are motivated by cases corresponding to scenarios where the correlation is positive in all environments, but it is unstable. These theoretical settings are motivated by real world “shortcut learning” problems (e.g. [4], [5] among others). In our experiment we consider the popular Waterbirds example where it also holds that the background is predictive of the label in both environments.
>
> 3. While we agree that constructing algorithms with a similar guarantee to Algorithm 1 is not hard, we argue that this has to do with giving up on interpolation rather than using “hidden knowledge.” In our main result we show that no interpolating algorithm can be robust, even if they explicitly try optimize our knowledge of robust test error and use “hidden knowledge” such as what environment every training example is coming from, the values of $r_s,r_c,\theta_1,\theta_2$ and $\sigma$.
>
> We would be happy to learn whether the reviewer’s criticism lies mainly in their dissatisfaction with the invariant learning setting, or with the details of our results, so we can better accommodate a fruitful discussion going forward.
>
> [1] Peters, J., Bühlmann, P., & Meinshausen, N. (2016). Causal inference by using invariant prediction: identification and confidence intervals. Journal of the Royal Statistical Society: Series B (Statistical Methodology), 78(5), 947-1012.
>
> [2] Arjovsky, M., Bottou, L., Gulrajani, I., & Lopez-Paz, D. (2019). Invariant risk minimization. arXiv preprint arXiv:1907.02893.
>
> [3] Aubin, B., Słowik, A., Arjovsky, M., Bottou, L., & Lopez-Paz, D. (2021). Linear unit-tests for invariance discovery. arXiv preprint arXiv:2102.10867.
>
> [4] Geirhos, R., Jacobsen, J. H., Michaelis, C., Zemel, R., Brendel, W., Bethge, M., & Wichmann, F. A. (2020). Shortcut learning in deep neural networks. Nature Machine Intelligence, 2(11), 665-673.
>
> [5] Zech, J. R., Badgeley, M. A., Liu, M., Costa, A. B., Titano, J. J., & Oermann, E. K. (2018). Variable generalization performance of a deep learning model to detect pneumonia in chest radiographs: a cross-sectional study. PLoS medicine, 15(11), e1002683.

---

> ### Author Response · Authors · 2022-11-16
> **Paper Revision Uploaded, Further Response Would be Appreciated**
>
> We have uploaded a revision of the paper (please see [general comment](https://openreview.net/forum?id=dQNL7Zsta3&noteId=gE9MW_UlR7m)).
> We would be grateful if you consider our previous response and the revision, and let us know whether they address your concerns regarding the significance of our results.
>
> Thank you again for your comments and effort in reviewing our paper.

---

### Official Review · Reviewer_8afp · 2022-10-28

**Confidence:** 4
**Correctness:** 3
**Technical Novelty And Significance:** 3
**Empirical Novelty And Significance:** 3
**Recommendation:** 6

**Clarity, Quality, Novelty And Reproducibility:**

Clarity and Quality: The paper is relatively easy to understand. However, I think adding a figure illustrating the linear two environments in a picture may help reader understand the problem setting much faster.

Novelty: I am not very familiar with all relevant literature and to what extent the model setting here is reasonable / too simplified to all the researchers in the field. The paper does seem self-explanatory though.

Reproducibility: My best guess is that the work should be reproducible.

**Strength And Weaknesses:**

Strength:
The idea of the paper is neat. The claims are in general well-supported.

Weaknesses:
1. For Theorem 1, why is the probability over the draw S? while the robust error is a deterministic function of w? I guess the S here refers to the space of possible N samples, not the set of data points, but please clarify.
2. Please also clarify on the drawing of \mu_s and \mu_c in Part 3 of Theorem 1. The paper mentioned that both are uniformly distributed on a sphere, but they also need to be orthogonal -- is \mu_s drawn from the subspace that is orthogonal to \mu_c? Also, 65 in Theorem 1 seems to be a strange number, please clarify as well.

**Summary Of The Paper:**

The paper proposes a neat observation for a simple linear overparameterized classifier for a simple "linear two environment problem": interpolating classifier and invariance property can not be achieved at the same time. The paper also presents an algorithm that is provably invariant in this simple setting while being a non-interpolating classifier.

**Summary Of The Review:**

I would vote for marginally above the acceptance threshold, given that there are minor issues that need to clarify (see above), and I am not in the best position to comment on the novelty aspect, although I find the paper self-explanatory.

---

> ### Author Response · Authors · 2022-11-11
> **Author Response**
>
> We thank the reviewer for appreciating the idea of the paper, for recognizing its novelty, and for making the effort to fully understand the paper without being familiar with all the related work. To address the weaknesses pointed out by the reviewer:
>
> 1. $S$ is indeed drawn from the space of possible $N$ samples. The robust error is a deterministic function of $w$ once we fix $\mu_c$ and $\mu_s$. After these are fixed, we require a concentration result over the $N$ sample as in standard learning settings. We hope this clarifies this technical detail.
>
> 2. Related to the above, $\mu_s$ is indeed drawn from the subspace orthogonal to $\mu_c$. This is described more precisely in Appendix A where we define notation and describe the data generating process in more detail; in the revision we will further clarify this point in the main text. We thank the reviewer for this comment and hope that it improves the manuscript.
> The reason we opt for orthogonality between the features is that it enables a linear classifier to depend only on the invariant feature and discard the spurious one completely.
>
> 3. As for the requirement $N > 65$ in Theorem 1, this is a technical detail that we were not able to avoid due to the Equal Opportunity constraint. To be able to evaluate this metric reliably from an $N$-sample we must have (with high probability) some minimal number of examples with label $Y=1$. This is stated very briefly in footnote 5, but we will make sure to better clarify the role of this requirement in the statement of Theorem . For the proof of Theorem 1, we simply plug the desired success probability (0.99) into the expression described above.
>
> **Adding a figure to describe the setting.** Taking the reviewer’s suggestion regarding an additional figure describing the setting, we will add such a figure to Section 2. We appreciate this suggestion and hope that it improves clarity.
>
> **Adequacy of the model and technical novelty of the result.** Finally, to better clarify the novelty and simplicity of the setting w.r.t prior works, we note that this model has been considered in prior works on related topics ([1] is perhaps the closest to our particular setting, but also [2] is quite close). Therefore we think it is an acceptable model to study some aspects of problems regarding spurious correlations and fairness. While we agree that it is simple, its analysis can be challenging, and in terms of technical novelty our results considerably extend those of [1] where results are given on the max-margin classifier (instead of all interpolating classifiers considered in this work). We have expanded the discussion on novelty w.r.t previous work in the related work section.
>
> [1] Sagawa, S., Raghunathan, A., Koh, P. W., & Liang, P. (2020, November). An investigation of why overparameterization exacerbates spurious correlations. In International Conference on Machine Learning (pp. 8346-8356). PMLR.
>
> [2] Rosenfeld, E., Ravikumar, P. K., & Risteski, A. (2020, September). The Risks of Invariant Risk Minimization. In International Conference on Learning Representations.

---

> ### Author Response · Authors · 2022-11-16
> **Paper Revision Uploaded, Further Response Would be Appreciated**
>
> We have uploaded a revision of the paper (please see [general comment](https://openreview.net/forum?id=dQNL7Zsta3&noteId=gE9MW_UlR7m)). We would be grateful if you consider our previous response and the revision, and let us know whether there are any concerns regarding our work that are not addressed by them.
>
> Thank you again for your helpful comments and effort in reviewing our paper.

---

### Official Review · Reviewer_aRur · 2022-10-29

**Confidence:** 3
**Correctness:** 3
**Technical Novelty And Significance:** 3
**Empirical Novelty And Significance:** 2
**Recommendation:** 5

**Clarity, Quality, Novelty And Reproducibility:**

The work was fairly clear and, from what I am aware, novel.  There were a few points that could use additional clarity/clarification:

(C1) Theorem statement: presumably this should be gamma >, not gamma <, since it says 'lower bound'.  For part 3 of the theorem, presumably this gamma is not the same as the gamma in the theorem statement, since in the subsequent paragraphs you say that the result holds for classifiers with 'arbitrarily small margin gamma'.  I am assuming this refers to *only part 3 of the theorem*, since the theorem statement presumably requires a lower bound for gamma.    I was also confused if gamma was supposed to play a role in part 2 of the theorem -otherwise, it doesn't seem to appear anywhere?

**Strength And Weaknesses:**


Understanding the benefits and pitfalls of interpolation is an important research question.  It is worthwhile to try and develop insights by the study of simpler models, like the linear models considered in this work, and to then translate and verify these intuitions into more complex neural network models.  I quite liked the authors' attempt at studying overparameterization by using ResNet baseline features that are then mapped into a higher-dimensional space via random features.  This provides a novel way of introducing overparameterization that is attractive due to its ease of computation.  I'm unsure if this approach had been introduced before (can the authors comment?), but I liked it.

However, I found serious problems with the framing and interpretation of the setup and of the results in the paper.  In particular, the claim that "interpolation is at odds with invariance" does not seem to be supported by the theoretical setup they showed.  In the proof, they consider theta_1=1 and theta_2=0, so that the source distribution is a mixture of two components: (1) Gaussian mixture with mean y(mu_c + mu_s), and (2) Gaussian mixture with mean y mu_c.  They assume that ||mu_s|| >> ||mu_c||, and show that an interpolator on the source data will rely more upon mu_s than upon mu_c, and that this causes the "robust error" to be large because any reliance upon mu_s increases the "robust error".  There are a number of problems with this setup:

(W1) If ||mu_s|| >> ||mu_c||, it isn't clear to me why we should think of mu_s as "spurious": it is the dominant feature for data that comes from N( y(mu_c + mu_s), sigma^2 I).  And as the authors mention in the second-to-last paragraph of Section 2, the relationship of ||mu_s|| to ||mu_c|| matters much more than the number of samples from each environment in the overparameterized regime, so large ||mu_s|| really is the determining factor.  Because of this, I don't believe mu_s is really a 'spurious' feature: it is the most important feature for the setting considered.

(W2) The definition of "robust error" seems odd in light of point (1).  The error gets worse if the learner relies on mu_s at all, but this doesn't make sense if we think of mu_s as an important feature.

**Summary Of The Paper:**

The authors consider the generalization of interpolating models under distribution shift by focusing on a linear classification problem.  They consider a source distribution that can be interpreted as a `two environment' distribution, and define a notion of `robust error' that measures generalization on a (family of) target distribution(s).    They show that there exist learning rules which can achieve good robust error, but any interpolating rule will have bad robust error.  They verify their results with experiments.  However, I believe there are issues with their setup that I am not sure can be fixed easily.

**Summary Of The Review:**

~~I think some of the central claims in the paper are not supported by the evidence provided by their work.  In particular, the theory setup relies upon a mixture of source distributions and the notion of a "spurious" feature, but the setup seems to imply that this feature is not spurious but is actually the dominant, useful feature.  If I have mis-understood something, I am happy to reconsider my opinion.  ~~


==== post discussion with other reviewers ====
I met with other reviewers and the AC to discuss the paper.

On the one hand, my fundamental criticisms still stand: if one considers a distribution with a large core feature and a small spurious feature, most natural algorithms will fail to be invariant in the sense the authors describe, and it appears previous work has indeed shown that in similar settings natural ERM-based algorithms will fail to be invariant.  Thus, it is not surprising that interpolating models will exhibit the same failure to invariance, and it isn't clear what insight this provides into the phenomenon of interpolation: previous work on benign overfitting (and classical statistical learning theory) has established how brittle interpolating models can be even for within-distribution generalization.

On the other hand, it appears some machine learning practitioners may have taken from the 'benign overfitting' results that interpolation is not only something that can be sometimes ok, but is something that should be encouraged.  This work then clearly shows that, in at least the setting of high-dimensional gaussian class-conditional distribution with large  spurious features, interpolation precludes invariance.

In summary, although I believe this work is borderline, I would be fine if the AC/PC accepts the paper.

---

> ### Author Response · Authors · 2022-11-11
> **Author Response**
>
> We thank the reviewer for the effort put into reading our paper and hope the following will clarify some points while convincing the reviewer that the setting is both interesting and relies on a widely acceptable model in the literature.
>
> Since we are eager to make the paper accessible to readers who are not fully familiar with all the literature we build upon, we would be grateful if the reviewer comments on whether the clarifications below are useful in reconsidering their opinion about the setup.
>
> We respond to the review’s main concerns (W1 and W2) with the following points.
>
> **Robustness to a strong spurious signal is well motivated by prior work.** Our setting relies on many previous works that study learning in the presence of spurious correlations, or fairness requirements. Specifically, the case where $\| \mu_s \| > \| \mu_c \|$ yet we still want to avoid relying on  $\mu_s$ models the Waterbirds scenario (where $\mu_s$ is the background type and $\mu_c$ is the bird type). This setting is used in our experiments but is also studied in many prior works ([1, 2, 3] to name a few). This formal setting can also depict the ColoredMNIST problem from IRM [4] (where $\mu_s$ would correspond to color and $\mu_c$ to digit), which was used in many papers afterwards (there it also holds that both $\theta_1$ and $\theta_2$ are positive and we would still like the classifier to discard $\mu_s$). As for other purely synthetic settings, Examples 2 and 3 of [5] propose similar settings to study properties of invariant learning algorithms, where the spurious feature can be learned with smaller weights, or induces a larger margin. Furthermore the worst case error (robust error in our paper) is a standard measure in all of these settings. In fact, many papers consider the robust error with respect to unbounded shifts (e.g. the min-max risk over all possible environments in IRM, see also [6, 7] and many other works that consider worst case error on possible distribution shifts), while we limit $\theta$ to the $[-1, 1]$ interval to accommodate for finite sample results and study a setting that is not as extreme as unbounded shifts.
>
> **The spurious signal does not have to be much stronger than the core signal.** We would like to emphasize that our result does not require that $\| \mu_s \| \gg \| \mu_c \|$ as the reviewer suggested. While we do require that $\| \mu_s \|$ is larger than $\| \mu_c \|$, this is not by an order of magnitude, as the quantity $\sqrt{N_2}/(N_1 \gamma)$ does not necessarily diverge when the sample size grows. In our simulations the ratio $\mu_s / \mu_c$ is simply set to $2$, which seems to us like a rather small and reasonable ratio to consider. In the revision, we will expand the discussion on the regime of parameters where the result holds.
>
> **Our results continue to hold for negative $\theta_2$ values.** We note that it is also possible to prove the result when the correlation in one environment has opposite sign to the other environment (e.g. $\theta_1=1$ and $0 > \theta_2 > -1$, when $N_2 / (N_1^2 \gamma)$ is sufficiently small). Our choice of $\theta_2=0$ is a matter of convenience and readability, as it simplifies the terms in the proof.
> Since a negative value of $\theta_2$ better motivates our notion of robust error, we will gladly extend our proofs to that setting in the revision.
>
> Finally, we clarify point C1 raised in the review.
>
> **The role of $\gamma$ in Theorem 1.** The parameter $\gamma$ denotes the smallest margin our results are valid with respect to - the smaller $\lambda$ is, the stronger the statement. Theorem 1 allows setting $\gamma$ arbitrarily small while at the same time guaranteeing that a margin of at least $1/(4\sqrt{N})$ is achievable. Therefore, the guarantee of the Theorem is non-vacuous as long as $\gamma <1/(4\sqrt{N})$, which is why the inequality appears in the Theorem statement. If this is confusing, we can simply remove it in the revision.
>
> [1] Sagawa, S., Koh, P., Hashimoto, T.B., & Liang, P. (2020). Distributionally Robust Neural Networks. ICLR.
> [2] Puli, A. M., Zhang, L. H., Oermann, E. K., & Ranganath, R. (2021, September). Out-of-distribution Generalization in the Presence of Nuisance-Induced Spurious Correlations. ICLR.
> [3] Idrissi, B. Y., Arjovsky, M., Pezeshki, M., & Lopez-Paz, D. (2022, June). Simple data balancing achieves competitive worst-group-accuracy. CLeaR (pp. 336-351). PMLR.
> [4] Arjovsky, M., Bottou, L., Gulrajani, I., & Lopez-Paz, D. (2019). Invariant risk minimization. arXiv preprint arXiv:1907.02893.
> [5] Aubin, B., Słowik, A., Arjovsky, M., Bottou, L., & Lopez-Paz, D. (2021). Linear unit-tests for invariance discovery. arXiv preprint arXiv:2102.10867.
> [6] Kamath, P., Tangella, A., Sutherland, D., & Srebro, N. (2021, March). Does invariant risk minimization capture invariance?. AISTATS (pp. 4069-4077). PMLR.
> [7] Rosenfeld, E., Ravikumar, P. K., & Risteski, A. (2020, September). The Risks of Invariant Risk Minimization. ICLR.

---

> > ### Comment · Reviewer_aRur · 2022-11-16
> > **Response to response**
> >
> > Thanks for the detailed response.   I re-examined the papers linked in the response.  I can see that in some previous work, authors have used the word "spurious" to refer to signals that are significantly stronger than "invariant" signals.  I would have the same critique for these works if I had reviewed them, as I am not convinced it is reasonable to refer to signals which are the most useful and dominate all other signals as "spurious".
> >
> > In my view, it is not accurate to claim that "interpolation is fundamentally at odds with invariance" based on the facts the authors have presented: namely, there exists a family of distributions where in Distribution 1, there is a high-signal feature and a low-signal feature, and interpolators will perform well in-distribution by relying on the high-signal feature, while if you shift to Distribution 2 where the low-signal features dominate, then the interpolator will fail to generalize.  I think it is still an interesting observation that for the distribution the authors consider, interpolators will not learn a diverse set of (more transferable) features when a single strong and simple feature suffices for learning in-distribution.  (Although it is worth noting that previous theoretical work on neural networks trained by gradient descent [has shown a similar phenomenon](https://proceedings.neurips.cc/paper/2020/hash/6cfe0e6127fa25df2a0ef2ae1067d915-Abstract.html)).   Still, the authors do not make this rather weaker claim, but make the much stronger claim that they have exposed a fundamental tension between interpolation and invariance.
> >
> >
> > Regarding the claim about $\theta_2 < 0$.  As the authors have not yet posted a revised version, it is hard to assess this claim, and the revision deadline is in quite soon.  But my understanding (please correct me if I am wrong) is that requiring $\sqrt{N_2}/(N_1 \gamma)$ to be smaller requires that $|\mu_s|$ to be larger than $|\mu_c|$ by Assumption (3) in Proposition 1.  This brings us to the same problem that I brought up in my initial review and that I re-iterated above, i.e. the "spurious" feature is actually the most dominant feature.

---

> > > ### Author Response · Authors · 2022-11-16
> > > **Continued Discussion and Notification That the Revision is Available**
> > >
> > > Thank you for re-examining the works mentioned in our response and for clarifying the main concerns regarding our work. We have uploaded a revision, which includes the extended statement for different values of $\theta_2$. In the regime where $\sqrt{N_2}/(N_1\gamma)$ is small, our results holds for $\theta_2 <0$ *and* for $r_c$ in the same order of magnitude as $r_s$, since in this case the RHS in the final condition in Eq. (3) is a constant.
> > >
> > > To the best of our understanding, there are two main concerns in your review. The first lies in the use of the term “spurious” and perhaps in the formal setup this leads to, while the second is in our use of the term “fundamentally at odds”. We further respond on these points below.
> > >
> > > **The statement that “interpolation is at odds with invariance”.** Following your comments and those of reviewer VPKR, we realized that these terms may be perceived differently from how we intended and consequently changed the title of the paper to say that “interpolation can provably preclude invariance”. We have made corresponding edits in other places of the paper that mention the words “at odds”. We hope you find that this better reflects our results.
> > >
> > > **Adequacy of the formal setting and the use of the word “spurious.”** We would like to suggest distinguishing the concerns regarding the use of the term “spurious” and those regarding the formal setting and our results. While the term “spurious” might be interpreted differently depending on the reader, our formal use of this term is well defined and clear.
> > > As you observe in your response, there are works that use the term “spurious correlation” in the same way we do in our paper - some of them well cited and accepted to ICLR in previous years. We also suggest that this use is common in works on the topic, and we are very far from being the first to motivate highly similar formal settings to ours using this language. This is not to argue whether this use of language is correct or not, but rather to suggest that this is an issue to be resolved at the community-level. Understanding whether the term “spurious” is warranted will require extensive study of real-world scenarios and how they are reflected in formal settings. Clearly, this is outside the scope of our paper.
> > > We think that when evaluating our work, the appropriateness of the term “spurious” should not be conflated with whether the formal setting we study is interesting, and whether the formal results are novel and correct.
> > >
> > > **Evaluation of the reviewer.** As noted above, while we see that there is some level of disagreement between our views and interpretation of the problem, we think that the evaluation of our work which states that “Several of the paper’s claims are incorrect or not well-supported.” is quite harsh. As far as we understand from your comments, you do agree that the results are novel, and the formal claims are proved. Hence, regardless of our disagreements, it seems plausible to agree that at the very least our formal results are correct, they did not appear in previous work, and the model we consider has been studied in many prior works hence the results should be of interest to a significant part of the community. We would be grateful if you consider this in your final evaluation, or indicate where we are wrong in our understanding.
> > >
> > > Thank you again for your further engagement in the discussion.

---

> > > > ### Comment · Reviewer_aRur · 2022-11-18
> > > > **response**
> > > >
> > > > Thanks for your comments.  I think the new title is an improvement.  The main reason for my previous evaluation that "Several of the paper’s claims are incorrect or not well-supported" was due to the overclaiming of results ("fundamentally at odds" is unwarranted, as the authors concede), but the title change has gone some way to assuage my concerns.  I am raising my score on this evaluation, as well as the overall evaluation from reject (3) to weak reject (5).
> > > >
> > > > My evaluation is that, if one accepts the formal setup without judgement, your results are correct and new.   My main issue is with the formal setup.  And this difficulty comes from my perception that there is not much insight that comes from the observation that for a distribution where there is a large signal component and a small signal component, a certain class of learning algorithms will favor the large signal component, and then by definition will not work well when considering a different setting where the large signal component is absent or degraded.  It seems likely that many learning algorithms will "preclude invariance" for this particular distribution in high-dimensional settings (using the authors' definition of "preclude invariance").  It is interesting that any interpolation algorithm will fail in this setting, but it seems many non-interpolating algorithms would have the same behavior.  If the authors were able to show that 'standard' non-interpolating learning algorithms did not preclude invariance (and not just the manually crafted Algorithm 1), this would provide more solid evidence that interpolation is responsible for precluding invariance, rather than the imbalance of the signal between "spurious" and "core" features.
> > > >
> > > > I acknowledge the authors' point that at least 2 previous works have made similar assumptions.  I don't believe this precludes the ability of future reviewers to take issue with these assumptions. The cited works are all within the last 3 years, and for such recent work (and even older work) I think it's reasonable to debate whether the assumptions are reasonable and allow for significant, novel insights into fundamental aspects of interpolation learning.  I am not yet convinced this is the case for this work, which is the reason for my score, but it is reasonable to disagree with my assessment.

---

### Official Review · Reviewer_7Zqk · 2022-11-04

**Confidence:** 3
**Correctness:** 3
**Technical Novelty And Significance:** 3
**Empirical Novelty And Significance:** 3
**Recommendation:** 8

**Clarity, Quality, Novelty And Reproducibility:**

## Unsurprising that there exist problems where interpolation and invariance are at odds

Consider, for simplicity, rotating your problem so that $\mu_c = r_c e_1 = (r_c, 0, \dots, 0)$ and $\mu_s = r_s e_2 = (0, r_s, 0, \dots, 0)$.
Then your data distribution is
$$
y \sim \operatorname{Unif}\\{-1, 1\\}
\qquad
x \mid y \sim \mathcal N\left( \begin{bmatrix}y r_c \\\\\ \theta y r_s \\\\ 0 \\\\ \vdots \\\\ 0 \end{bmatrix}, \sigma^2 I_d \right)
\tag{A}
.$$
In the main motivating setup of your paper, you'd like there to exist a large-margin separator for this problem, and moreover for it to exhibit benign overfitting, i.e. to have low generalization error.

In the main motivating setup, you'd also like for there to _not_ be an invariant large-margin separator, which is exactly the question of there not existing a large-margin separator for the $(d-1)$-dimensional problem
$$
y \sim \operatorname{Unif}\\{-1, 1\\}
\qquad
\tilde x \mid y \sim \mathcal N\left( \begin{bmatrix}y r_c \\\\ 0 \\\\ \vdots \\\\ 0 \end{bmatrix}, \sigma^2 I_{d-1} \right)
\tag{B}
.$$

That this situation can occur is a large part of the results of your paper (the other part being everything about Algorithm 1).
Intuitively, though, it's clear when framed like this that this is possible – if (A) is "right on the boundary" of having a separator of a given margin, and $\theta r_s$ isn't too small, then (B) won't.

(I don't know offhand of previous results that establish the achievable margin for (A) or (B), but they're _almost_ addressed by the far more general results of section 6 of the (concurrent) paper [Zhou et al. (2022)](https://arxiv.org/abs/2210.12082), who only study Gaussian $x$ but I believe can be extended to handle $x$ being a mixture of a small number of Gaussians fairly straightforwardly, and then show both a margin bound and conditions for benign overfitting in the squared hinge loss.)

All of this is only about the existence of any high-margin separator. Maybe this was in e.g. Rosenfeld et al. (2020) or Nagarajan et al. (2020) and I don’t remember (in which case you should recap the results here), but it would also be good to understand what particular algorithms – especially the max-margin separator – do in situations like this one: how much weight do they put on $\mu_s$? We'd expect that even when there exist invariant interpolators, the max-margin separator will still put a lot of weight in the "bad" direction $\mu_s$ (since it doesn't know it's bad).

## Experimental results

For the simulations, the experiments are fine and show that your algorithm works in the case it was designed for, although it'd be nice to also see maybe the test accuracy vs $d$ for a value of $\theta$ or two. I'm also especially wondering whether having a $\theta = 0$ environment is very important to the qualitative nature of your results, since it will presumably put very little weight towards $\mu_s$, and then the fair learning step will likely mostly just use that predictor since it's naturally going to be nearly fair. Does something similar happen if you have a $\theta = 1$ and $\theta = 0.5$ environment?

For Waterbirds: these results do indeed show that the test FNR gap is decreased by the regularizer even for high $d$, which is good! The amount of improvement there is fairly small, though, and the problem somewhat contrived. I'd be interested to know if this holds up if there are more than two training environments, if the individual-environment predictors are not themselves linear, and so on.

Also, almost all of the improvement in test FNR gap is achieved by "Ours ($\lambda = 0)$", indicating that the vast majority of the added invariance of your method comes from combining predictors learned on each environment separately. (I'm not aware of this scheme having been used previously in invariant learning, but it's closely related to some schemes in meta-learning-type areas, particularly [CAVIA](https://arxiv.org/abs/1810.03642) if only the "late-layer" parameters are task-specific, and also multiple kernel learning where the kernels are themselves learned on related tasks as in [MetaMKL](https://arxiv.org/abs/2106.07636).) To understand this a little more, it'd be good to also see results for just $v = (1, 0)$ and $v = (0, 1)$, i.e. just using one or the other predictor, for test FNR in both environments, compared to the outcome of the fair learning.

## Miscellaneous small questions

- Why use $\lVert v \rVert_\infty = 1$ as the constraint in the second phase of Algorithm 1? Taking a convex combination of the $w$s might would feel more natural, but it also isn't obvious that any constraint at all is needed, other than I suppose inside the analysis.

## Typos, etc

- There are several uses of \citet that should be \citep, e.g. at the bottom of the first page.
- Section 5.1 setup: "we further fix $r_c =1$ and $r_c = 2$" – presumably the second one should be $r_s$ :)
- Many papers in your bibliography are cited as arXiv papers but were in fact published at conferences, e.g. I noticed Rosenfeld et al. (2020) and Nagarajan et al. (2020) which were both published at ICLR 2021. A tool like [rebiber](https://github.com/yuchenlin/rebiber) could catch all of those for you.

**Strength And Weaknesses:**

Strengths:

- The explicit incompatibility between interpolation and robustness in a specific setting is good to have worked out.
- The fact that Algorithm 1 works on one particular problem setting is also good to know.
- Based on one experiment, the algorithm seems like it might be somewhat effective in (linear) interpolating regimes, where previous invariance approaches such as IRM totally fail.

Weaknesses:

- The parts of the main theoretical result that aren't about the algorithm seem to be unsurprising; see below.
- This means Algorithm 1 is relatively more important to the overall quality of the paper. But Algorithm 1 is a little bit underexplored:
  - It's only defined for two environments (though there's an obvious extension to more)
  - It's only theoretically analyzed in one quite-restrictive setting
  - It's only empirically analyzed for:
    - a small number of instances of that restrictive setting,
    - and one relatively-unrealistic real-data experiment, where empirical results do show some ability to do _something_ in the interpolating regime, but at quite a high cost (see below).

**Summary Of The Paper:**

This paper proves that, in a particular class of two-environment linear models containing both relevant and irrelevant directions, interpolation is incompatible with invariance (having low "robust error" as used in out-of-domain generalization, or also corresponding to fairness constraints).

It also proposes an algorithm, based on learning a fair/robust linear combination of predictors learned independently on each environment, can provably learn a predictor with low robust error, and shows that it works reasonably well both in the setting to which the theoretical bounds apply and also for a linear model on pretrained features for Waterbirds, where – unlike previous approaches – it does (slightly) reduce the FNR gap past the interpolation threshold.

**Summary Of The Review:**

This paper has two main contributions: one theoretically establishing a conflict between interpolation and invariance, and one suggesting an algorithm and showing it works in this theoretical setting as well as some very limited empirical results.

The first contribution is unsurprising; working out the details is valuable, but I'm not sure the way the paper is currently written that casual readers will come away with the right idea about "how often" this holds, etc.

The second is more surprising, but somewhat under-evaluated, as described above.

Overall, I think the point of the paper is important – interpolating settings are major flaws in current approaches to invariant learning, as has been noted before e.g. in the discussion of the IRM paper – but I'm somewhat unhappy with its framing and the thoroughness of its exploration and evaluation. I'm giving the paper a 6 because I think it matters and I think it's above the minimal bar for ICLR, but there's a hypothetical version of this paper with some more work that I would be far more excited about.

---

> ### Author Response · Authors · 2022-11-11
> **Author Response (1/2)**
>
> We thank the reviewer for the careful examination of the results and suggestion for alternative derivations. We also appreciate the reviewer’s recognition of the value of our results and would like to clarify points that we think are crucial to take into account when critiquing our work. We will lay them out below, where in our main claim we hope to clarify why the result in the first part of our paper is far from being trivial.
>
> ### Why our main result is nontrivial.
>
> The reviewer suggests that the first part of our result is not surprising because one may arrange the problem parameters such that invariant classifiers do not exist. We address this concern in three parts: first we explain why such an inexistence approach cannot prove Theorem 1, second we explain how our proof actually works (unlearnability rather than inexistence), and finally we argue that the research preceding our result is evidence for its nontrivial nature.
>
> 1. **Theorem 1 requires a robust interpolator to exist.** Note that the second part of Theorem 1 implies that the Bayes-optimal robust (and invariant) classifier $\mathbf{w}=\mu_c$ has robust error at most $\epsilon$. Taking $\epsilon$ to be sufficiently small (e.g., $\epsilon = N^{-5}$) therefore guarantees that with very high probability (e.g., at least $1-N^{-4} > 0.9999\%$) $\mathbf{w}=\mu_c$ does interpolate the training data. Furthermore, a short calculation (which we will provide in the revision) shows that it does so with a decent margin of $\Omega(1/\sqrt{N})$. Consequently, setting the model parameters such that a robust interpolator does not exist is incompatible with the statement of Theorem 1.
>
> 2. **How we prove our result.** To show that all interpolating algorithms fail to find an invariant interpolator (that exists!), we leverage the high-dimensional nature of the problem to show that---based on the training data---no interpolating classifier can distinguish the invariant interpolating direction from the infinitely many other interpolating directions (that have comparable or better margin). **That is, while an invariant interpolator exists, it is unlearnable from the finite training data.** The proof is technically involved and leverages concentration of measure on the sphere and convex duality - we are not aware of a simpler argument for showing this result. Note also that for proving such a result we must let $\mu_c$ and $\mu_s$ be random vectors, since otherwise a “learning” algorithm that always outputs $\mathbf{w}=\mu_c$ would be successful.
>
> 3. Why our result is nontrivial. First, a number of recent works [1,2,3] attempt to come up with interpolating learning rules that are invariant and/or robust to spurious features in overparameterized settings, and our result highlights a basic limitation of such approaches. Second, previous works [4,5] have attempted to show limitations of interpolating learning rules, but have only been able to argue about the max-margin classifier, as opposed to any interpolator. Finally, we reached our result only after months of looking for an interpolating learning rule that probably works in the setting we consider, so it certainly surprised us!
>
> We thank the reviewer for bringing up this important point, which was not sufficiently explained in our original manuscript. **In the revision** we will soon submit, we will include a detailed discussion of this point as well as a simulation demonstrating existence (as opposed to learnability) of invariant interpolators.

---

> > ### Author Response · Authors · 2022-11-11
> > **Author Response (2/2)**
> >
> > **The choice of $\theta_2$ is not crucial.** We set $\theta_2=0$ for convenience, as it simplifies many of the terms in the proof. However, there is no inherent difficulty in extending the results to all $\theta_2 > -1$. In the revision we will provide such theoretical extension (when $N_2 / (N_1^2 \gamma)$ is sufficiently small) as well as supporting simulations.
> >
> > **Answers regarding Waterbirds experiment.** While we agree that the setting is not representative of a real-world scenario, we believe that a thorough investigation of the variety of non-interpolating techniques on real-world data is better kept for future work. We chose to focus on a specific question, that has quite a few intricate technical details, and as we state in the paper we do not see Algorithm 1 as the main contribution here. Nonetheless, we did check some of the questions raised by the reviewer. Since the environments in Waterbirds are not balanced in terms of labels (water background means that most labels are of waterbirds, while land background is the opposite), it leads the individual models for each environment in the Waterbirds dataset (i.e. v = (1, 0) and v = (0, 1)) to have low FNR (since one is mostly correct when Y=1 and the other is mostly correct when Y=0). But this also yields very low accuracy (since they predict poorly on images from the other environment). Then the additional linear classifier on top of the concatenated representation (with $\lambda=0$) improves accuracy considerably, while $\lambda > 0$ allows for further trade-off between FNR and accuracy. We also note that even though the improvement in FNR for $\lambda>0$ is small compared to the gap from interpolating models, it is expected to see smaller gains over an FNR that is already relatively low.
> >
> > We thank the reviewer again for their effort, it is clear that a lot of thought went into the review.
> >
> > [1] Kini, G. R., Paraskevas, O., Oymak, S., & Thrampoulidis, C. (2021). Label-imbalanced and group-sensitive classification under overparameterization. Advances in Neural Information Processing Systems, 34, 18970-18983.
> > [2] Wang, K. A., Chatterji, N. S., Haque, S., & Hashimoto, T. (2021). Is Importance Weighting Incompatible with Interpolating Classifiers?. arXiv preprint arXiv:2112.12986.
> > [3] Arjovsky, M., Bottou, L., Gulrajani, I., & Lopez-Paz, D. (2019). Invariant risk minimization. arXiv preprint arXiv:1907.02893.
> > [4] Sagawa, S., Raghunathan, A., Koh, P. W., & Liang, P. (2020, November). An investigation of why overparameterization exacerbates spurious correlations. In International Conference on Machine Learning (pp. 8346-8356). PMLR.
> > [5] Nagarajan, V., Andreassen, A., & Neyshabur, B. (2020, September). Understanding the failure modes of out-of-distribution generalization. In International Conference on Learning Representations.

---

> > > ### Comment · Reviewer_7Zqk · 2022-11-13
> > > **Thanks**
> > >
> > > Thanks for your responses. I look forward to seeing the revision, since your answers here have clarified some key points (and expect to increase my score).

---

> > > > ### Author Response · Authors · 2022-11-16
> > > > **Thank You for Responding, Revision Has Been Uploaded**
> > > >
> > > > Thank you very much for continuously engaging in the discussion and for your helpful comments.
> > > >
> > > > We are glad that our response clarified key points, and hope that the revision we have just uploaded addresses these points appropriately.

---

### Public Comment · ~LIN_Yong1 · 2022-11-07
**Dicussion with Related Work**

This is an interesting paper on discussing the overparameterization and invariance. We'd like to have a discussion with on related works [1] and [2] on this topic.

**On the results**:
*  To the best of our knowledge, the confict between overparamterization and invariance learning is first formally identified in [1]. [1] shows that any overparameterized model that can interpolate the data can fail the invariance constraint.
* [2] analyzes a linear regression case. [2] shows that a model that can interpolate data always achieves a smaller IRM loss than the model merely using invariant features (IRM loss with zero penalty is standard ERM loss).

------------update--------------

Thanks for Reviewer 7Zqk and author's discussion and correction.

It seems that we have some misunderstanding on the work before. This paper presents the results that:  the max margin interpolator will rely on the spurious features with large probability. Whereas,  [1, 2] is about the existence of a interpolator that fails the IRM constraint which still uses the spurious feature.  So they typically focus on different aspects. The results in this paper bring new insights.



[1] Yong Lin, Hanze Dong, Hao Wang, Tong Zhang; Bayesian Invariant Risk Minimization; CVPR 2022

[2] Xiao Zhou, Yong Lin, Weizhong Zhang, Tong Zhang; Sparse Invariant Risk Minimization. ICML 2022

---

> ### Comment · Reviewer_7Zqk · 2022-11-07
> **Slight correction**
>
> Hi,
>
> (A reviewer here.) It’s not the case that this model requires there to be only informative or spurious features and nothing else; rather, it assumes there’s an informative direction, a single orthogonal spurious direction, and then $d-2$ remaining directions that are neither, but these directions might not be axis-aligned and aren’t known by the learner. An individual feature dimension in the input space might contain components of all three kinds.
>
> Personally, I think the rephrasing of the problem as (A) in [my review](https://openreview.net/forum?id=dQNL7Zsta3&noteId=BoUkc8PUku5) makes the scope of the setup clearer. (I absolutely agree it is not particularly realistic; in particular, a slightly more realistic setting might allow for multiple spurious directions that change separately across environments rather than a single linear index, whose interaction with interpolation might then be more interesting.)

---

> > ### Public Comment · ~LIN_Yong1 · 2022-11-08
> > **Thanks for the correction.**
> >
> > Thanks for discussion and correction.
> >
> > It seems that we have some misunderstanding on the setting. Thank you for pointing it out. And we finally get your point. We will revisit this part after.
> >
> > We have a question on another (possible more general and simple) setting: Let us consider $n$ samples, each sample $x$ contains invariant features $x_v \in \mathbb{R}^{d_v}$, $x_s \in \mathbb{R}^{d_s}$ and $x_r \in \mathbb{R}^{d_r}$. For simplicity, we just assume there is no feature transformation: $x=[x_v, x_s, x_r]$. $x_v$ and $x_s$ are informative of $y$. $x_r$ are just independent random Gaussian noise, i.e., $x_r \sim \mathcal{N}(0, I)$. If the random features are high dimensional, i.e., $d_v + d_r >> n$, we can certainly interpolate the $n$ samples by only using  invariant features $x_v$ and random feature $x_r$. Will the result in  Theorem 1 (point 3) "the robust error of $w$ is at least 1/2"(must use the spurious feature) still holds in this setting?

---

> > > ### Comment · Reviewer_7Zqk · 2022-11-08
> > > **Only if carefully designed to**
> > >
> > > There are two questions: does there exist a robust interpolator? If so, does any particular algorithm, e.g. min-norm interpolation = max-margin separator, find a robust interpolator?
> > >
> > > There will be a robust interpolator if interpolation is possible while ignoring $x_s$; if $d_r$ is big enough, this will be possible. In the regression setting, an interpolator will a.s. exist as long as $d_v + d_r \ge n$. The classification problem in the paper, though, is set up such that a large-margin separator does not exist unless you use $x_s$.
> > >
> > > For the second question, the min-norm interpolator will only have “benign overfitting” at all under some conditions; one prominent one is if $x_v$ has much bigger variance than the $x_r$ (spiked covariance). Most of the time, it will likely also use $x_s$ significantly, because it doesn’t know not to; Nagarajan et al. (2020) looked at this type of problem.

---

> > > > ### Public Comment · ~LIN_Yong1 · 2022-11-10
> > > > **What if we only have low dimensional $x_v$ and $x_s$ and no $x_r$**
> > > >
> > > > Thank you so much. We really appreciate your comment.

---

> ### Author Response · Authors · 2022-11-11
> **Author Response**
>
> Thank you for reaching out for discussion. Below, we discuss the reference [1,2] pointed out in the comment in detail. We also address your comment regarding other settings where interpolating models might succeed, and point out a possible misunderstanding regarding our main claim and its proof.
>
> As for the points raised:
>
> * Note that both [1] and [2] in the comment draw conclusions that are specific to the IRMv1 and/or VREx regularizers and are not general claims about all interpolating algorithms. We also cited [2] when discussing works that showed the empirical failure of some interpolating algorithms in learning invariant models.
>
> * To our best understanding, the claim in [1] is specific to IRMv1 with 0-1 loss as opposed to the more often used log-loss. In addition to proving a limitation of any interpolating classifier, in Appendix G of the revised paper we analyze the implicit bias of linear classification on separable data with IRMv1 and the log-loss, showing that it corresponds to max-margin classification. The revision will also compare our results to [1].
>
> * As mentioned in the comment, in [2] the result is given for regression, which is not the setting we consider here. More importantly, the proposition in the paper claims that a representation containing all features can achieve lower loss than the ground truth invariant representation. It does not discuss whether an interpolating algorithm can somehow prefer a classifier on top of that representation that uses only invariant features, or whether it will learn a representation that contains the invariant features or not. Hence it does not answer the question we ask in this paper, even for specific regularizers.
>
> * Regarding the setting, please note that our model has added Gaussian noise (i.e. it is not just two features as claimed in the initial comment). While we never claimed this setting depicts a complete real world scenario, it is the nature of theoretical work to consider simplified cases that are amenable to analysis and capture some important aspects of real world problems. We believe that any algorithm dealing with a far more complex real world scenario, should be able to solve this stylized setting. Furthermore, this type of stylized setting is considered in many previous works that study aspects of learning in the presence of spurious correlations, or minority groups ([SRKL20, RRR20] and several other important works in the field). We would be happy to be advised regarding which claims of the paper the commenter perceives as misleading or untrue. Finally - and perhaps most importantly - **our claim is not about the inexistence of an interpolating solution that is invariant, it is rather about whether an interpolating learning rule can learn such a model** (see also response to reviewer 7Zqk). More specifically, you are correct in pointing out that an invariant model may exist which uses the invariant feature and noise to fit the data perfectly (see also our added simulation in appendix H of the revision). However, that does not mean an interpolating learning rule can **find** this solution. We show that when $\mu_c$ and $\mu_s$ are randomized (i.e. they cannot be hard-coded into the learning rule), there is a large regime of parameters where no interpolating learning rule can retrieve this solution.
>
> [SRKL20] Sagawa, S., Raghunathan, A., Koh, P. W., & Liang, P. (2020, November). An investigation of why overparameterization exacerbates spurious correlations. In International Conference on Machine Learning (pp. 8346-8356). PMLR.
> [RRR20] Rosenfeld, E., Ravikumar, P. K., & Risteski, A. (2020, September). The Risks of Invariant Risk Minimization. In International Conference on Learning Representations.

---

### Author Response · Authors · 2022-11-11
**Author Response to All Reviewers**

We thank all the reviewers for their great effort and insightful reviews - they will certainly help improve clarity and presentation of our paper. We are also glad that all reviewers are in agreement about the novelty of our result, and that most reviewers found it interesting and valuable.

The reviewers have raised several concerns, and we believe that our responses fully address them. Nevertheless we are looking forward to a fruitful discussion, and to respond to any other comments that may arise. Below we provide short answers to comments that appeared in more than one review, and we provide detailed answers to each reviewer in the dedicated threads.

1. **Existence of an invariant interpolator.** Both reviewer 7Zqk and Lin Yong suggested that our main result boils down to constructing a setting where no invariant predictor can interpolate the data. As we explain in detail in our response to reviewer 7Zqk, this is not how our proof works and moreover such strategy cannot prove our main result, which holds for arbitrarily small robust test (and training) error. In the model we consider, **invariant interpolators exist but are not learnable from the data** due to information-theoretic barriers to learning in the presence of high-dimensional noise.

2. **Spurious vs. core signal strength.** Reviewers p2tP and VPKR were concerned that our results only hold when the spurious signal is much stronger than the core signal. However, Theorem 1 as well as our simulations actually show the fragility of interpolating learning  rules in regimes (often studied in prior work) where the spurious signal is only slightly stronger than the core signals, and such regimes were often studied in prior work.

3. **The value of $\theta_2$.** Another common concern is that Proposition 2 considers the setting $\theta_1=1$ and $\theta_2=0$, so that our robust error metric unfairly requires learning algorithms to anticipate the possibility of test environments with negative $\theta_2$. However, the choice $\theta_2=0$ was mainly a mathematical convenience: we can extend our proofs to all $\theta_2\in{[-1,0]}$ straightforwardly (but with a substantial increase in length) when $N_2$ is sufficiently small compared to other problem parameters).

4. **The fundamental nature of our setting.** Lin Yong, as well as Reviewers p2tP and VPKR note that our result holds for a specific data model, and question whether interpolating learning might successfully learn invariant classifiers in other models. While it is true that our setting is specific, it is also **extensively studied** and **arguably the simplest** setting in which one can study classification under spurious correlations. Therefore, in our opinion, algorithms that do not reliably succeed in this fundamental linear classification setting are fundamentally flawed as invariant learning rules, and we have little confidence they will work in more complex settings.

In the revision - that we will soon submit - we will address all of these points in detail. In particular, we will discuss at length how we show non-learnability (as opposed to inexistence) of robust classifiers, and we will extend our results to any $\theta_2 \ge -1$.

We thank the reviewers again and look forward to hearing their thoughts after reading the response.

---

### Author Response · Authors · 2022-11-16
**Paper Revision Uploaded, Further Comments Are Welcome**

We have uploaded a revision of the paper, with the main changes listed below. In the revision, edits addressing key concerns from the reviewers are highlighted in blue.
1. We changed the title of the paper to more concretely describe the scope of the results. The new title is “Malign Overfitting: Interpolation Can Provably Preclude Invariance.” We believe that this reflects the result quite well, and hope that it addresses some of the concerns of reviewers aRur and VPKR.

2. We have modified the proof and statement of Proposition 1 and Theorem 1 to accommodate different values of $\theta_2$. The change in the statement is a condition that $\theta_2 > - \frac{N_1 \gamma}{\sqrt{288 N_2}}$. Hence $\theta_2$ can be set to low values (e.g. -1), while still ruling out the learning of invariant models by interpolating rules, for sufficiently high $N_1^2 \gamma^2 / N_2$. In the appendix we include an extra term in Eq. (3) of the statement, showing that positive values of $\theta_2$ result in a wider range of parameters where interpolating algorithms fail. This term was omitted from the main text for better readability. Results of an additional simulation with $\theta_2=-\frac{1}{2}$ (other parameters are as in the previous simulation) can also be found in the appendix.

3. We have added Appendix G where a formal result (in the style of [Rosset et. al. 03]) regarding the inductive bias of IRMv1 is given, along with a simulation showing the cosine similarity between linear models learned by a few algorithms and the one learned by SVM. This strengthens the claims made in the paragraph after Proposition 1, which were supported by our simulations, and also relates to the references mentioned by Lin Yong (we also discuss the reference in the appendix).

4. The point regarding proof of un-learnability (our result) vs. inexistence, that appeared in the discussion with reviewer 7Zqk and Lin Yong, is emphasized in the introduction and after the statement of Theorem 1. We have also added a calculation and simulation regarding invariant interpolators in Appendix C.1, and changed the term “interpolating classifier” in the abstract to “interpolating learning rule,” to better avoid confusion between the two.

5. We have added a plot to Appendix A (not in the main text, due to space constraints) that visualizes the analyzed setting, as suggested by reviewer 8afp, along with a reference in the main text.

6. In Section 6 we have added discussion regarding prior work that analyzes two-staged algorithms at the infinite-data limit, brought up by reviewer VPKR. In case there are additional points that the reviewer thinks are important here, we will work to incorporate them either before the revision deadline, or in a later version of the paper.

We sincerely thank all reviewers for highly insightful commentary, which we believe has led to substantial improvements in the paper. We would be happy to make any additional edits and engage with further comments to our best capabilities in the remaining discussion period.

[Rosset et. al. 03] Rosset, S., Zhu, J., & Hastie, T. (2003). Margin maximizing loss functions. Advances in neural information processing systems, 16.

---

### Decision · Program_Chairs · 2023-01-20

**Decision:**

Accept: poster

**Justification For Why Not Higher Score:**

Results are not surprising although the framing of the paper suggests otherwise, some overclaiming that should be addressed

**Justification For Why Not Lower Score:**

Consensus amongst reviewers that the problem is interesting and could lead to additional theory and applications down the line

**Metareview: Summary, Strengths And Weaknesses:**

This is an interesting paper that is not without notable flaws, as flagged by a majority of the reviewers. It sits at the intersection of two relevant contemporary topics: Benign overfitting and invariant learning. The reviewers consisted of experts on both sides.

On the one hand, the results are not at all surprising. There is no reason to expect interpolating classifiers to be invariant, and benign overfitting is a very specific phenomenon that is not believed to hold widely. In the discussion, reviewers agreed on this point. Moreover, the original title of the paper clearly oversold its contributions, and while the revised title ("Malign Overfitting: Interpolation Can Provably Preclude Invariance") is an improvement, it might be better to avoid the connection with "benign overfitting". Overall, the paper would still benefit from tempering its claims; here are two such examples:

- The results are not "_even_ in the simplest of settings", but in fact in a very specific setting. It is not clear if any of this generalizes, and if it does, this should be formalized.
- The authors do not show that "interpolating models are _fundamentally_ less invariant than non-interpolating ones", but again, only in a very specific setting.

On the other hand, the paper accomplishes two important goals, although they have may not been the originally intended goals: 1) It provides an interesting perspective on two important problems, which will hopefully lead to fruitful dialogue between the communities, and 2) Proposition 1 on the information-theoretic impossibility of interpolators to be invariant seems to be novel and a nice contribution to the literature. Again, reviewers agreed on these points.

Ultimately, although some reviewers expressed valid reservations, after discussion there was a weak consensus to accept this paper. Ideally, the authors will temper their claims and qualify their results with the fact that benign overfitting is not expected to hold universally.

**Note From Pc:**

if the above contains the word "oral" or "spotlight" please see: "oral" presentation means -> notable-top-5% and "spotlight" means -> notable-top-25%. As stated in our emails, we are disassociating presentation type from AC recommendations

**Summary Of Ac-Reviewer Meeting:**

See above.